

# Ensemble-based observation impact of surface $CO_2$ concentration observations on analysis and forecast of atmospheric $CO_2$ concentrations over East Asia

Min-Gyung Seo[1] and Hyun Mee Kim[1]

[1]Atmospheric Predictability and Data Assimilation Laboratory, Department of Atmospheric Sciences, Yonsei University, 50 Yonsei-ro, Seodaemun-gu, Seoul, 03722, Republic of Korea

*Correspondence to*: Hyun Mee Kim (khm@yonsei.ac.kr)

**Abstract.** The effect of assimilated surface $CO_2$ concentration observations on the analysis and forecast errors of model $CO_2$ concentrations was evaluated using the data assimilation (DA)-forecast system combining the Weather Research and

Forecasting model coupled with Chemistry and modified Data Assimilation Research Testbed. To investigate the impact of surface $CO_2$ observations, four observing system simulation experiments (OSSEs) were conducted in July 2019. The impact of $CO_2$ observations on the $CO_2$ concentration analysis was calculated using self-sensitivity. Average self-sensitivity of four OSSEs was 21.0%, implying that surface $CO_2$ observations provided average 21.0% information to the $CO_2$ concentration analysis. Self-sensitivity was highly correlated with the root mean square error of the analysis and hourly variability in the

surface $CO_2$ observations at each observation site. The impact of $CO_2$ observations on reducing forecast errors, calculated using nonlinear forecast error reduction (NER), showed that NER with DA was reduced by average 17.0% compared with that without DA. Linear forecast error reduction, calculated using the ensemble forecast sensitivity to observation (EFSO) impact, showed that the EFSO impact was greater at surface $CO_2$ observation sites with higher self-sensitivity and active vegetation types. Average fraction of beneficial observations for all experiments was 68.9% (66.3%) for 6 h (12 h) forecasts,

implying that more than half of the assimilated $CO_2$ observations contributed to reducing forecast errors. The assessment of observation impact on the $CO_2$ concentration analysis and forecast can be useful for monitoring and estimating atmospheric $CO_2$ concentrations, optimizing surface $CO_2$ fluxes, and designing atmospheric $CO_2$ observation networks.

## 1 Introduction

Atmospheric $CO_2$ concentrations are influenced by various factors, including $CO_2$ emissions from several sources,

respiration and photosynthesis of vegetation, and interactions between the atmosphere and the oceans (Ussiri and Lal, 2017). Efforts have been made to reduce the recently elevated atmospheric $CO_2$ concentrations (Le Quéré et al., 2019; Zhang et al., 2020). To verify whether efforts to reduce $CO_2$ emissions have been implemented, the distribution of the surface carbon flux has to be accurately estimated. Many studies have attempted to optimize surface carbon fluxes using inverse modeling by combining atmospheric chemistry models with data assimilation (DA) (Kim et al., 2014a, 2014b; Kim et al., 2017a; Kim et





al., 2018; Monteil et al., 2020; Park and Kim, 2020; Wang et al., 2020; Maksyutov et al., 2021; Zhang et al., 2021; Cho and Kim, 2022). In previous studies that optimized the surface carbon flux and atmospheric $CO_2$ concentration using modeling, surface carbon flux observations were not used, but atmospheric $CO_2$ concentration observations were used in the DA. Atmospheric $CO_2$ concentration observations include surface in situ and flask observations, aircraft observations, and satellite observations. The characteristics of the $CO_2$ concentration observations from various sources differ (Byrne et al.,

2017). In particular, the number of $CO_2$ concentration observations in East Asia, which is important for understanding the carbon cycle, is smaller than those in North America and Europe (Byrne et al., 2017; Seo and Kim, 2023; Seo et al., 2024a, b). Therefore, understanding how each observation reduces the analysis and forecast errors of $CO_2$ concentration estimation is crucial for optimizing atmospheric $CO_2$ concentrations using DA with limited atmospheric $CO_2$ concentration observations. Using the Lorenz 40-variable model and ensemble Kalman filter (EnKF) DA, Liu et al. (2009) showed that the impact of

individual observations on analysis can be calculated using self-sensitivity, which is the diagonal component of influence matrix. By applying the influence matrix presented in Liu et al. (2009) to CarbonTracker, Kim et al. (2014b) showed that the self-sensitivity (i.e., analysis sensitivity) of surface $CO_2$ observations is greater in Asia, where surface $CO_2$ observations are sparse. Kim et al. (2017a) showed that, when Japan–Russia Siberian Tall Tower Inland Observation Network (JR-STATION) observations were additionally assimilated into CarbonTracker, the observation impact of these added JR-STATION

observations was approximately 60% greater than that of North American tower observations. Cho and Kim (2022) showed that the observation impacts of Anmyeon-do (AMY) and Gosan (GSN) (11.71% and 11.38%, respectively) were greater than the globally averaged observation impact (6.14%) when the $CO_2$ concentration observations from AMY and GSN located in Korea were additionally assimilated into CarbonTracker. Therefore, the impact of each $CO_2$ concentration observation used for DA on the $CO_2$ concentration analysis can be quantitatively calculated using self-sensitivity.

Impact of each $CO_2$ concentration observation used for DA on $CO_2$ concentration forecasts can also be analyzed by observing system experiments (OSEs; Jung et al., 2010; Lawrence et al., 2019; Laroche and Poan, 2022; Kim and Kim, 2023; Ratheesh et al., 2023; Martin et al., 2023) and forecast sensitivity observation impact (FSOI; Jung et al., 2013; Kim et al., 2017b; Mallick et al., 2017; Kim and Kim, 2018a; Kim and Kim, 2019; Kotsuki et al., 2019; Privé et al., 2021; Kim and Kim, 2021; Daescu and Langland, 2022). An OSE analyzes the impact of individual observations by conducting DA experiments,

with each observation denied. To evaluate the impact of N observations, N number of experiments needs to be conducted, which necessitate expensive computational resources (Jung et al., 2010; Jung et al., 2013; Kim et al., 2017b; Kim and Kim, 2018b; Kim and Kim, 2021). Adjoint-based FSOI analyzes the impact of individual observations using backward adjoint integration without conducting N experiments. However, one limitation is that an adjoint model is necessary to calculate the observation impact on forecast errors. Liu and Kalnay (2008) and Kalnay et al. (2012) presented methods for calculating the

FSO based on ensembles (i.e., ensemble forecast sensitivity to observation; EFSO) without an adjoint model. Observation impact analysis using EFSO has been used in several studies. By assimilating meteorological observations using the National Centers for Environmental Prediction (NCEP) Global Forecast System (GFS) EnKF system, Ota et al. (2013) showed that most observations reduced forecast errors and the impact of satellite radiance observations was the greatest. By analyzing the



observation impact using the Weather Research and Forecasting (WRF) model and the local ensemble transform Kalman filter (LETKF), Casaretto et al. (2023) showed that all observations, except surface pressure observations, reduced forecast errors in southeastern South America. Using the observation impact based on EFSO in the Climate Forecast System (CFS) v2 - LETKF system, which is an air–sea coupled modeling and DA system, Chang et al. (2023) showed that the forecast performance of sea temperature and salinity was improved by excluding observations with negative impacts. Using WRF and Gridpoint Statistical Interpolation (GSI)-based EnKF, Gasperoni et al. (2024) showed that EFSO can be applied to a high-resolution regional modeling system by analyzing the impact of radar observations in addition to conventional observations in the Dallas–Fort Worth testbed region. However, no studies have analyzed the impact of $CO_2$ observations on reducing errors in $CO_2$ concentration forecasts using the EFSO.

In this study, the impacts of surface $CO_2$ concentration observations on analysis after DA and on $CO_2$ concentration forecasts were investigated to better analyze and forecast atmospheric $CO_2$ concentrations in East Asia. As the number of surface $CO_2$ observations is small in East Asia, the observation impacts of pseudo $CO_2$ concentration observations from added and redistributed surface $CO_2$ observation sites, as well as those of pseudo observations from existing surface $CO_2$ observation sites, were evaluated by conducting an observing system simulation experiment (OSSE). Because OSSE extracts pseudo observations from simulation results which are assumed to be the true state, evaluating the observation impact in the desired observation locations is feasible. Therefore, the impact of each observation on the analysis and forecast was examined by assimilating the pseudo surface $CO_2$ observations extracted from various observation networks. Section 2 presents the model and data assimilation, the method for calculating the observation impact, OSSE, and the experimental framework. Section 3 presents the results, and Section 4 presents the summary and conclusions.

## 2. Methods

### 2.1 Model and data assimilation

WRF coupled with Chemistry (WRF-Chem) and Data Assimilation Research Testbed (DART) system were used to assimilate surface $CO_2$ concentration observations and forecast surface $CO_2$ concentrations. The WRF-Chem is an atmospheric chemistry transport model developed by the National Center for Atmospheric Research (NCAR). In WRF-Chem, the transport processes of chemical species were simulated by the input emission data and simulated meteorology. As the $CO_2$ concentration was treated as a tracer that was transported without atmospheric chemical reactions in WRF-Chem, only $CO_2$ concentrations were simulated in this study.

DART is a DA system developed at the NCAR that can be easily combined with various numerical weather prediction models (Anderson et al., 2009). To assimilate the surface $CO_2$ observations, the ensemble adjustment Kalman filter (EAKF) in DART was used. In this study, DART was modified to be combined with WRF-Chem: an observation operator for surface $CO_2$ concentration observations was added to assimilate surface $CO_2$ observations in DART, and some of the code within DART was modified accordingly (Seo and Kim, 2025).





## 2.2. Ensemble adjustment Kalman filter

EAKF, a type of EnKF, updates the ensemble mean ($\bar{\mathbf{x}}_a$) and ensemble member $(\mathbf{x}_a)_i$ of the analysis as follows:

$$\bar{\mathbf{x}}_a = \mathbf{P}^a[(\mathbf{P}^b)^{-1}\bar{\mathbf{x}}_b + \mathbf{H}^T\mathbf{R}^{-1}\mathbf{y}_o], \tag{1}$$

$$\mathbf{P}^a = [(\mathbf{P}^b)^{-1} + \mathbf{H}^T\mathbf{R}^{-1}\mathbf{H}]^{-1}, \tag{2}$$

$$(\mathbf{x}_a)_i = \mathbf{A}^T((\mathbf{x}_b)_i - \bar{\mathbf{x}}_b) + \bar{\mathbf{x}}_a, \ i = 1,\cdots,N, \tag{3}$$

where $\mathbf{P}^a$ is the analysis error covariance, $\mathbf{P}^b$ is the background error covariance, $\bar{\mathbf{x}}_b$ is the ensemble mean of the background, $\mathbf{H}$ is the linearized observation operator, $\mathbf{R}$ is the observation error covariance, $\mathbf{y}_o$ is the observation, and $(\mathbf{x}_b)_i$ is the $i$th ensemble member of the background. $\mathbf{A}$ in Eq. (3) satisfies the relationship in Eq. (4).

$$\mathbf{P}^a = \mathbf{A}\mathbf{P}^b\mathbf{A}^T. \tag{4}$$

The experimental settings for assimilating the surface $CO_2$ concentration using EAKF were as follows: As diurnal cycle of $CO_2$ concentration due to vegetation activities was clear, the assimilation window was ± 1 h and the cycling interval was 6 h. In previous studies that optimized $CO_2$ concentrations and carbon fluxes by assimilating $CO_2$ concentration observations, the cycling interval was 6 h or 24 h (Kang et al., 2011, 2012; Zhang et al., 2021; Liu et al., 2019; Liu et al., 2022). The number of ensembles was set to 20. The method of Gaspari and Cohn (1999) was applied for the localization of surface $CO_2$

observations, with a localization radius of 1274.2 km. Initial perturbations were applied to the meteorological and chemical variables of 20 ensemble members. The initial perturbations of the meteorological variables were obtained from 150 perturbation banks using be.dat.cv3 of the WRF DA (WRFDA). Initial perturbations of the chemical variables (i.e., $CO_2$ concentrations) were produced using the method presented in Yumimoto (2013) and Miao (2014). Inflation was applied to maintain the ensemble spread during the forecast. Spatially varying state space inflation, based on the Gaussian distribution,

was applied as prior inflation (Anderson et al., 2009) and relaxation to prior spread (RTPS) was applied as posterior inflation (Whitaker and Hamill, 2012). The multiplicative inflation of the RTPS was 1.0. To maintain ensemble spread, multi physics options were also applied for microphysics, cumulus, and planetary boundary layer schemes when conducting ensemble forecasts. The physical parameterization schemes used for the ensemble forecasts are listed in Table 1.



**Table 1. Physics options used for WRF-Chem.**

| Shortwave and longwave radiation | | Rapid Radiative Transfer Model (RRTMG) scheme (Iacono et al. 2008) |
|---|---|---|
| Surface layer | | Revised MM5 scheme (Jiménez et al. 2012) |
| Land surface | | Unified Noah Land Surface Model (Tewari et al. 2004) |
| Microphysics | Single forecast | WRF Single-moment 6-class scheme (Hong and Lim 2006) |
| | Ensemble forecast | Purdue Lin scheme (Chen and Sun 2002) |
| | | WRF Single-moment 6-class scheme (Hong and Lim 2006) |
| | | Thompson scheme (Thompson et al. 2008) |
| Cumulus | Single forecast | Grell–Freitas Ensemble scheme (Grell and Freitas 2014) |
| | Ensemble forecast | Kain–Fritsch scheme (Kain 2004) |
| | | Grell–Freitas Ensemble scheme (Grell and Freitas 2014) |
| | | Grell–Devenyi Ensemble scheme (Grell and Dévényi 2002) |
| Planetary boundary layer | Single forecast | Yonsei University (YSU) scheme (Hong et al. 2006) |
| | Ensemble forecast | Yonsei University (YSU) scheme (Hong et al. 2006) |
| | | Mellor–Yamada–Janjic (MYJ) scheme (Janjić 1994) |
| | | Mellor–Yamada Nakanishi Niino (MYNN) Level2.5 scheme (Nakanishi and Niino 2006) |

## 2.3. Observation impact

### 2.3.1. Observation impact to $CO_2$ concentration analysis (Influence matrix)

The impact of the surface $CO_2$ observations used for DA on the $CO_2$ concentration analysis was calculated using the

sensitivity of the analysis to the observations ($\mathbf{S}^o$) (i.e., influence matrix) (Liu et al., 2009; Kim et al., 2014b; Kim et al., 2017a; Park and Kim, 2020; Cho and Kim, 2022) as follows:

$$\mathbf{S}^o = \frac{\partial \mathbf{y}_a}{\partial \mathbf{y}_o} = \mathbf{K}^T \mathbf{H}^T = \mathbf{R}^{-1} \mathbf{H} \mathbf{P}^a \mathbf{H}^T, \tag{5}$$

where $\mathbf{y}_a$ is the analysis in the observation space and $\mathbf{K}$ is the Kalman gain. The diagonal component of the influence matrix is called the self-sensitivity. Eq. (5) can be converted into Eq. (6) in the EnKF DA.

$$\mathbf{S}^o = \mathbf{R}^{-1} \mathbf{H} \mathbf{P}^a \mathbf{H}^T = \frac{1}{k-1} \mathbf{R}^{-1} (\mathbf{H} \mathbf{X}_a)(\mathbf{H} \mathbf{X}_a)^T, \tag{6}$$

where $\mathbf{H} \mathbf{X}_a$ represents the analysis ensemble perturbation in the observation space and $k$ is the number of ensembles. In Eq. (6), $\mathbf{S}^o$ is proportional to the analysis error covariance and inversely proportional to the observation error covariance. If the observation errors are not correlated with each other, the diagonal component of $\mathbf{S}^o$ (i.e., self-sensitivity) can be calculated as Eq. (7), with a value between 0 and 1 (Cardinali et al., 2004).




$$S_{jj} = \frac{\partial (y_a)_j}{\partial (y_o)_j} = \left(\frac{1}{k-1}\right)\frac{1}{(\sigma_j)^2}\sum_{i=1}^{k}[(\mathbf{HX}_a)_i]_j \times [(\mathbf{HX}_a)_i]_j, \tag{7}$$

where $\sigma_j^2$ is the observation error variance of the $j$th observation.

### 2.3.2. Observation impact to CO₂ concentration forecast (Ensemble Forecast Sensitivity to Observation)

The observation impact of surface $CO_2$ observations used for the DA on $CO_2$ concentration forecasts was calculated using EFSO.

Forecast error is calculated as follows:

$$e = (\mathbf{x}^f - \mathbf{x}_t)^{\mathbf{T}}\mathbf{C}(\mathbf{x}^f - \mathbf{x}_t), \tag{8}$$

where $\mathbf{x}^f$ is the forecast, $\mathbf{x}_t$ is the true state, and $\mathbf{C}$ is the positive definite matrix that defines the norm. The forecast errors can be subdivided into two types:

$$e_a = (\mathbf{x}_a^f - \mathbf{x}_t)^{\mathbf{T}}\mathbf{C}(\mathbf{x}_a^f - \mathbf{x}_t), \tag{9}$$

$$e_b = (\mathbf{x}_b^f - \mathbf{x}_t)^{\mathbf{T}}\mathbf{C}(\mathbf{x}_b^f - \mathbf{x}_t), \tag{10}$$

where $\mathbf{x}_a^f$ is the forecast from the analysis, $\mathbf{x}_b^f$ is the forecast from the background, $e_a$ is the forecast error with DA, and $e_b$ is the forecast error without DA (Fig. 1). Nonlinear forecast error reduction (NER) is calculated as

$$\text{NER} = \Delta e = e_a - e_b. \tag{11}$$

NER is the difference in forecast error due to the assimilated observations to make analysis; thus, it is treated as a nonlinear observation impact or forecast error reduction by the impact of observations. A negative (positive) NER indicates that the assimilated observations reduce (increase) the forecast error. Eq. (11) can be converted into Eq. (13) by applying the linear assumption of Eq. (12) (Kalnay et al., 2012; Ota et al., 2013; Hotta, 2014).

$$\mathbf{MX}_a \approx \mathbf{X}_a^f, \tag{12}$$

$$\Delta e \approx \frac{1}{k-1}\delta \mathbf{y}_0^{\mathbf{T}}\mathbf{R}^{-1}\mathbf{Y}_a \mathbf{X}_a^{f\mathbf{T}}\mathbf{C}((\mathbf{x}_a^f - \mathbf{x}_t) + (\mathbf{x}_b^f - \mathbf{x}_t)), \tag{13}$$

where $\mathbf{M}$ is the tangent linear model of the nonlinear NWP model, $\mathbf{X}_a$ is the ensemble perturbation of analysis, $\mathbf{X}_a^f$ is the ensemble perturbation of forecast integrated from the analysis, $\mathbf{Y}_a$ is the ensemble perturbation of analysis in observation space, satisfying the relationship of $\mathbf{Y}_a \approx \mathbf{HX}_a$ (Note Eq. (6)).

To prevent the filter divergence problems of EnKF in calculating the EFSO, localization was applied to each surface $CO_2$ observation in Eq. (13). The observation impact of the $j$th observation at $l$th grid point reflecting localization can be calculated as

$$(\Delta e)_{l,j} = \frac{1}{k-1}(\delta \mathbf{y}_0)_j[\rho_l \mathbf{R}^{-1}\mathbf{Y}_a(\mathbf{X}_a^{f\mathbf{T}})_l C_{ll}((\mathbf{x}_a^f - \mathbf{x}_t) + (\mathbf{x}_b^f - \mathbf{x}_t))_l]_j, \tag{14}$$



where $\rho$ is the localization function of the $j^{th}$ observation in $l^{th}$ grid point. Linear forecast error reduction (LER), the linearized observation impact (EFSO), was calculated by adding all the observation impacts of the $j^{th}$ observation in $l^{th}$ grid point in Eq. (14) as

$$\text{LER (or EFSO)} = \sum_j \sum_l (\Delta e)_{l,j} \tag{15}$$

By comparing the NER in Eq. (11) and LER in Eq. (15), the validity of the linear assumption was examined. If the LER is valid, then NER and LER should be similar.

As the assimilated observation type and model variables in the state vector were the same (i.e., $CO_2$ concentration), **C** matrix in Eq. (14) was the identity matrix. As the $CO_2$ concentration is transported by wind as the forecast progresses in WRF-Chem, the localization center of a specific observation should also shift as the forecast time increases. Kalnay et al. (2012) and Ota et al. (2013) used moving localization, in which the localization center moved as the forecast progressed. In this study, moving localization using the average wind simulated in WRF-Chem was applied to calculate the EFSO, which is the impact of each observation on forecast error reduction.

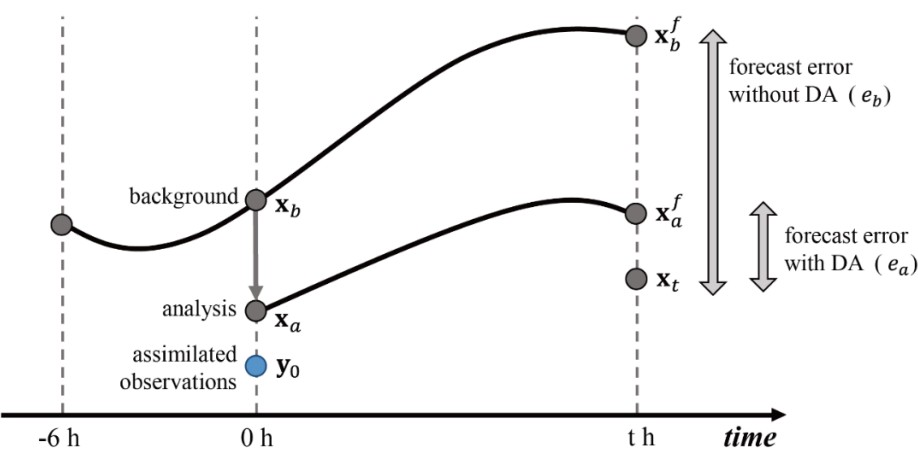

**Figure 1.** Schematic for calculating EFSO impact.

## 2.4. OSSE

### 2.4.1. Generation of true state and pseudo surface CO₂ observations

In OSSE, pseudo observations are extracted from the model simulation results, defined as the true state, and are used for DA. In this study, hourly $CO_2$ concentrations were simulated using WRF-Chem without DA and were used as the true state. The emission data used to simulate the true state were as follows: the average of CarbonTracker version 2022 (CT2022, https://dx.doi.org/10.25925/z1gj-3254) (Jacobson et al., 2023) and the Open-source Data Inventory for Anthropogenic $CO_2$ (ODIAC) v2020b (Oda and Maksyutov, 2015) for anthropogenic emissions, CT2022 (Jacobson et al., 2023) for oceanic



emissions, and the Vegetation Photosynthesis and Respiration Model (VPRM) (Mahadevan et al., 2008) built into WRF-Chem for biogenic emission. Hourly $CO_2$ concentrations, considered to be the true state, were generated by a single forecast. The physical parameterization scheme used for the single forecast is presented in Table 1. Final analysis (FNL) (NCEP/NOAA, 2000) was used as the meteorological initial and lateral boundary conditions, and CT2022 (Jacobson et al., 2023) was used as the chemical initial and lateral boundary conditions.

In OSSE, the true state must not deviate from the real natural variability (Masutani et al., 2010). The $CO_2$ variability of the true state in this study was included in the $CO_2$ variabilities of CT2022 and the Copernicus Atmosphere Monitoring Service (CAMS), which are widely used as $CO_2$ reanalysis (not shown). Figure 2 shows the distribution of the average surface $CO_2$ concentrations in the true state (Fig. 2a) and the distribution of the standard deviation of the hourly surface $CO_2$ concentrations in the true state (Fig. 2b). $CO_2$ concentrations were high in eastern and southern China, the western part of the Korean Peninsula, and near Tokyo, Japan. The standard deviation of hourly surface $CO_2$ concentrations was mainly large in areas with high $CO_2$ concentrations and northern China, where vegetation activity is high. The standard deviation of hourly surface $CO_2$ concentrations was closely related to the magnitude of daily variability in $CO_2$ concentrations. The regions where the $CO_2$ concentrations between day and night were quite different showed large standard deviations of hourly surface $CO_2$ concentrations.

Pseudo surface $CO_2$ concentration observations were generated using hourly $CO_2$ concentrations in the true state. After interpolating the true state $CO_2$ concentrations to the observation location, a Gaussian random observation error with a standard deviation of 1 ppm was added (Chen et al., 2023; Kang et al., 2011). All surface $CO_2$ observations used in this study were assumed to be of the in situ type with hourly $CO_2$ observations. In DA, the observation error variance of the extracted pseudo surface $CO_2$ observations was assigned a value of 2 ppm, as in Chen et al. (2023).

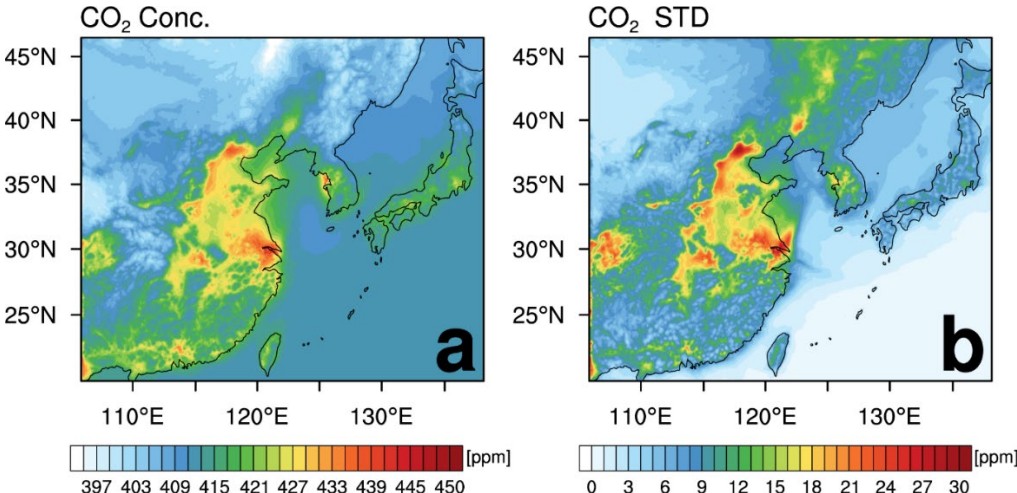

**Figure 2. Distribution of (a) average surface $CO_2$ concentrations [ppm] and (b) standard deviation of hourly surface $CO_2$ concentrations [ppm] in true state in the model domain.**



### 2.4.2. Strategies of selecting observation sites

Compared to North America and Europe, the number of surface $CO_2$ observations in East Asia is insufficient, and in situ observations are even fewer. Investigation of the observation impact using the existing in situ observations has limitations in fully understanding the characteristics of the observation impact. Therefore, to examine the impact of each observation used for DA on the simulated $CO_2$ concentrations under the framework of existing in situ surface observation sites and changed observation sites, four experiments with different observation networks were conducted in this study.

Two strategies (i.e., Random and Variability) were used to select surface $CO_2$ observation sites. The Random strategy randomly selects observation locations within the experimental domain, ensuring that the observation locations are at least 600 km from each other. This is because close observation locations do not provide the best results for reducing the forecast error (Yang et al., 2014). The Variability strategy selects observation locations from regions with highly variable true $CO_2$ concentrations, which have high standard deviation of the true state at each grid point (Fig. 2b). In the Variability strategy,

observation sites were selected starting from the grid with the greatest standard deviation, and the distances between the observation locations were at least 300 km from each other. For the Random and Variability strategies, the selected observation sites were located on land and at least 10 grids away from each domain boundary.

In all experiments (i.e., EXP1, EXP2, EXP3, and EXP4), the experimental settings were the same, except for the location of the surface $CO_2$ observations (Table 2). In EXP1, thirteen surface $CO_2$ observation sites were selected by Random strategy.

In EXP2, five surface $CO_2$ observation sites were World Data Centre for Greenhouse Gases (WDCGG) sites where hourly $CO_2$ observations were provided, and eight sites were in situ observation sites where hourly surface $CO_2$ observations were not provided by the WDCGG. Therefore, the observation sites used in EXP2 were the real observation sites. In EXP3, five surface $CO_2$ observation sites were WDCGG sites, and eight sites were selected by the Variability strategy. In EXP4, five surface $CO_2$ observation sites were WDCGG sites, three sites were selected by the Variability strategy, and five sites were

extracted by the Random strategy.

The locations of the surface $CO_2$ observation sites in each experiment are shown in Fig. 3 and Table 3. For observation sites selected by the Random (Variability) strategy, "R" ("V") preceded the site numbers.

The emission data used in the four experiments differed from those used to simulate the true state. This is to avoid the identical twin problem that can occur in the OSSE by setting the experimental design of the true state and four experiments

sufficiently different (Masutani et al., 2010; Shu et al., 2023; Kim et al., 2022). In the four experiments, ODIAC v2020b (Oda and Maksyutov, 2015) was used for anthropogenic emission, the Japan Meteorological Agency (JMA) ocean map (Iida et al., 2021; Takatani et al., 2014) was used for oceanic emission, and VPRM was used for biogenic emission.




**Table 2. Descriptions of the surface CO₂ observation sites in EXP1, EXP2, EXP3, and EXP4.**

| Exp. name | Description |
|---|---|
| EXP1 | 13 randomly distributed observation sites. |
| EXP2 | 8 in situ observation network sites and 5 WDCGG observation network sites. |
| EXP3 | 5 WDCGG observation network sites and 8 observation sites with high variability of hourly surface CO₂ concentrations. |
| EXP4 | 5 WDCGG observation network sites, 3 observation sites with high variability of hourly surface CO₂ concentrations, and 5 randomly distributed observation sites. |

**Table 3. Locations of the surface CO₂ observation sites in EXP1, EXP2, EXP3, and EXP4. The site ID and name were denoted when surface CO₂ observation sites exist.**

| EXP1 | | | EXP2 | | | EXP3 | | | EXP4 | | |
|---|---|---|---|---|---|---|---|---|---|---|---|
| Site ID | Lat [ºN] | Lon [ºE] | Site ID | Lat [ºN] | Lon [ºE] | Site ID | Lat [ºN] | Lon [ºE] | Site ID | Lat [ºN] | Lon [ºE] |
| R1 | 28.91 | 106.77 | YON (Yonagunijima, Japan) | 24.47 | 123.01 | YON | 24.47 | 123.01 | YON | 24.47 | 123.01 |
| R2 | 45.26 | 104.54 | AMY (Anmyeondo, Korea) | 36.54 | 126.33 | AMY | 36.54 | 126.33 | AMY | 36.54 | 126.33 |
| R3 | 40.47 | 123.26 | DDR (Mt. Dodaira, Japan) | 36.00 | 139.20 | DDR | 36.00 | 139.20 | DDR | 36.00 | 139.20 |
| R4 | 39.04 | 140.23 | KIS (Kisai, Japan) | 36.08 | 139.55 | KIS | 36.08 | 139.55 | KIS | 36.08 | 139.55 |
| R5 | 33.92 | 115.01 | RYO (Ryori, Japan) | 39.03 | 141.82 | RYO | 39.03 | 141.82 | RYO | 39.03 | 141.82 |
| R6 | 41.83 | 113.73 | XL (Xinglong, China) | 40.24 | 117.30 | V1 | 34.38 | 117.16 | V1 | 34.38 | 117.16 |
| R7 | 37.66 | 105.14 | HF (Hefei, China) | 31.90 | 117.17 | V2 | 40.19 | 111.32 | V2 | 40.19 | 111.32 |
| R8 | 46.58 | 120.60 | HKO (King Park, Hong Kong) | 22.31 | 114.17 | V3 | 29.50 | 105.03 | V3 | 29.50 | 105.03 |
| R9 | 26.05 | 113.28 | HKG (Hko Tsui, Hong Kong) | 22.21 | 114.25 | V4 | 38.48 | 114.82 | R14 | 23.74 | 108.80 |
| R10 | 35.56 | 128.05 | BO (Boseong, Korea) | 34.45 | 127.12 | V5 | 34.29 | 108.95 | R15 | 45.01 | 134.20 |
| R11 | 45.30 | 131.14 | JGS (Jeju Gosan Suwolbong, Korea) | 33.30 | 126.21 | V6 | 32.13 | 114.13 | R16 | 44.97 | 119.22 |
| R12 | 29.75 | 119.90 | ULD (Ulleungdo, Korea) | 37.48 | 130.90 | V7 | 23.00 | 112.10 | R17 | 29.95 | 112.69 |
| R13 | 24.11 | 121.27 | DOK (Dokdo, Korea) | 37.23 | 131.86 | V8 | 32.19 | 119.93 | R18 | 24.22 | 116.09 |





Figure 3. Distribution of surface $CO_2$ observation sites in (a) EXP1, (b) EXP2, (c) EXP3, and (d) EXP4. The randomly selected $CO_2$ observation sites (triangle); existing in situ $CO_2$ observation sites, with hourly $CO_2$ observations available from WDCGG (star); existing in situ $CO_2$ observation sites but hourly $CO_2$ observations are not provided (circle); and observation sites extracted from grids with large surface $CO_2$ variability (rhombus).

## 2.5. Experiments

FNL (NCEP/NOAA, 2000) was used for the meteorological initial and lateral boundary conditions, and CT2022 (Jacobson et al., 2023) was used for the chemical lateral boundary conditions. For the chemical initial condition, CT2022 was used only at the first time; thereafter, forecasts were conducted using the analysis produced by assimilating surface $CO_2$ observations





as the initial condition. Ensemble forecasts with 20 members were conducted in the $CO_2$ DA cycle, and the multi physical

parameterization schemes used in the ensemble forecasts are presented in Table 1.

The experimental period was from June 22 to July 31, 2019, and the spin up period for model stabilization was from June 22 to June 30, 2019. The experimental domain was East Asia with a horizontal resolution of 9 km and 51 vertical layers (Fig. 2). To sufficiently increase the ensemble spread to avoid filter divergence, a 12 h ensemble forecast was conducted only on June 22, 2019, at the beginning of the experiment, and the ensemble DA cycle was conducted every 6 h (00, 06, 12, and 18 UTC)

thereafter.

Self-sensitivity was calculated using analysis at 00, 06, 12, and 18 UTC. The EFSO was calculated for the 6, 12, 18, and 24 h forecasts at 00 UTC to avoid the influence of the diurnal cycle in $CO_2$ concentrations. To calculate the EFSO, 30 h and 24 h ensemble forecasts were conducted every 18 UTC (corresponding to – 6 h in Fig. 1) and 00 UTC (corresponding to 0 h in Fig. 1), respectively.

## 3. Results

### 3.1. Rank histogram

A rank histogram was used to check whether the ensemble forecasts were conducted appropriately in the ensemble DA-forecast system. The rank histogram was calculated by sorting each ensemble interpolated to the observation location and then counting the ranks corresponding to the observation values. If the shape of the rank histogram is flat, then the ensemble

spread is appropriate. If the rank histogram is U shaped, the ensemble spread is insufficient. If the rank histogram is dome shaped, the ensemble spread is excessive. The adjusted missing rate can be used as an indicator of the appropriateness of the ensemble spread (Hou et al., 2001; Meng and Zhang, 2008; Jung et al., 2012; Yang and Kim, 2021).

$$\text{Adjusted missing rate } = \left| \frac{2}{N_{ens}+1} - \text{ missing rate}\right|, \tag{16}$$

$$\text{Missing rate } = \text{ relative frequency of first rank } + \text{ relative frequency of last rank}, \tag{17}$$

where $N_{ens}$ is the total number of ensembles, which was 20 in this study. If the adjusted missing rate in Eq. (16) is less than 10%, the ensemble spread is appropriate.

Figure 4 shows a rank histogram of surface $CO_2$ concentrations for each experiment. Hamill (2001) showed that random observation noise should be added to each ensemble member for proper analysis of rank histogram. When an observation error is not considered, multiple observations can be counted in the first and last ranks. By adding random observation noise

to the observations, the ensemble system can be evaluated considering the total error, resulting in a flat rank histogram (Jung et al., 2012; Yang and Kim, 2021). Therefore, in this study, random observation noise was added to each ensemble member when calculating rank histograms. Rank histograms for all the four experiments were relatively flat (Fig. 4). In particular, the number of counts was uniform across all ranks in EXP1. The adjusted missing rate, which is an indicator that determines the appropriateness of ensemble spread, was 1.9%, 0.4%, 9.6%, and 8.1% for EXP1, EXP2, EXP3, and EXP4, respectively. The



adjusted missing rates in all four experiments were less than 10%, indicating an appropriate ensemble spread during the entire experimental period. In EXP3 and EXP4, more observations were counted at higher ranks (Figs. 4c and d). This was because the assimilated $CO_2$ concentrations in EXP3 and EXP4 were higher than the ensemble $CO_2$ concentrations, which resulted in a positive bias. Some of the assimilated $CO_2$ observations in EXP3 and EXP4 were selected using the Variability strategy, which selected observation sites from regions of high hourly variability in surface $CO_2$ concentrations simulated in the true state. As the pseudo surface $CO_2$ observations selected using the Variability strategy were often higher than the $CO_2$ concentrations of the ensemble (not shown), the rank histograms of EXP3 and EXP4 skewed to higher ranks. However, because the adjusted missing rates were all below 10% (9.6% and 8.1% for EXP3 and EXP4, respectively), the DA of the surface $CO_2$ observations was conducted appropriately in all four experiments.

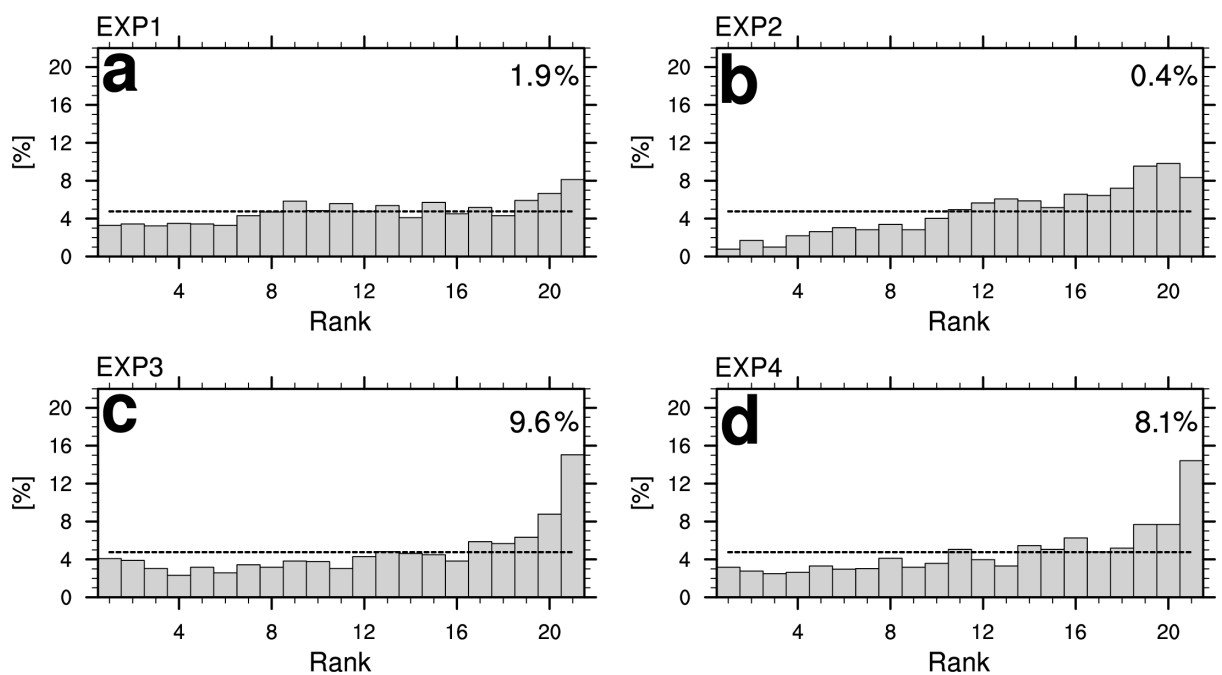

**Figure 4. Rank histogram for (a) EXP1, (b) EXP2, (c) EXP3, and (d) EXP4. The ideal value for rank histogram (e.g., 100%/21 = 4.8%) (dashed gray line). Number at the upper right corner in each figure represents the adjusted missing rate.**

## 3.2. Impact of surface $CO_2$ observations on analysis

Figure 5 shows the distribution of self-sensitivity for each observation site where the surface $CO_2$ observations were assimilated and Table 4 presents the self-sensitivity values for each observation site. The average self-sensitivity was 16.5%, 17.5%, 28.7%, and 21.3% for EXP1, EXP2, EXP3, and EXP4, respectively. As self-sensitivity represents the observation impact on the analysis generated in each assimilation cycle, the 16.5% self-sensitivity of EXP1 implies that 16.5% and 83.5%



information from observations and the background, respectively, were reflected in generating the analysis in the assimilation
cycle. In the four experiments, the average 21.0% and 79.0% information from the observations and the background, respectively, were reflected in the analysis. Among the four experiments, the impact of $CO_2$ observations on the analysis was the greatest in EXP3. The self-sensitivity was high at the observation sites in EXP3 and EXP4, where the surface $CO_2$ observation sites were selected based on the Variability strategy. In particular, the self-sensitivity of V1 site, where the hourly $CO_2$ variability was the greatest, was 52.2% in EXP3 and 51.1% in EXP4. In each assimilation cycle, more than half
of the observational information from V1 site was reflected in the analysis.

According to Liu et al. (2009), first calculated self-sensitivity in an ensemble DA system, the root mean square error (RMSE) of the analysis and self-sensitivity at observation sites are positively correlated, and self-sensitivity increases as the observation coverage becomes sparse for uniformly distributed observation sites. In Eq. (7), the self-sensitivity is proportional to the analysis error variance and inversely proportional to the observation error variance. As all surface $CO_2$
observations assimilated in this study were of the in situ type, the observation error variance at all observation sites was the same. Therefore, according to Eq. (7), the self-sensitivity is proportional to the analysis error variance.

As the number of assimilated surface $CO_2$ observation sites in the four experiments was the same as thirteen, the correlation between RMSE of the analysis (i.e., analysis error) and self-sensitivity was analyzed at each observation site (Fig. 6a). As self-sensitivity was high at observation sites with large hourly variability in surface $CO_2$ concentrations (Fig. 5), the
correlation between the hourly standard deviation of pseudo surface $CO_2$ observations and self-sensitivity was also analyzed (Fig. 6b). The greater the analysis error at each observation site, the greater the self-sensitivity (Fig. 6a). The correlation coefficient between the analysis error and self-sensitivity at each surface $CO_2$ observation site for all experiments was 0.84, indicating a strong positive correlation. Among the four experiments with DA, the analysis errors were greater in EXP3 and EXP4 than in the other experiments, resulting in greater self-sensitivity in EXP3 and EXP4, as shown in Fig. 5. The
correlation coefficient between the hourly standard deviation and self-sensitivity at each surface $CO_2$ observation site for all experiments was 0.87, indicating a strong positive correlation (Fig. 6b). As surface $CO_2$ observations were assimilated every 6 h, self-sensitivity was greater for observations located in regions with greater diurnal variations in $CO_2$ concentrations. When surface $CO_2$ observations were assimilated with a relatively short cycling interval of 6 h, self-sensitivity was closely correlated with the analysis error and hourly variability of observed $CO_2$ concentrations.






**Table 4. Average self-sensitivity [%] for each observation site in EXP1, EXP2, EXP3, and EXP4.**

| EXP1 | | EXP2 | | EXP3 | | EXP4 | |
|---|---|---|---|---|---|---|---|
| Site ID | Self-sensitivity | Site ID | Self-sensitivity | Site ID | Self-sensitivity | Site ID | Self-sensitivity |
| R1 | 23.8 | YON | 5.1 | YON | 5.0 | YON | 5.0 |
| R2 | 5.3 | AMY | 27.0 | AMY | 26.8 | AMY | 26.8 |
| R3 | 19.5 | DDR | 19.1 | DDR | 19.0 | DDR | 19.0 |
| R4 | 12.2 | KIS | 18.1 | KIS | 17.9 | KIS | 18.0 |
| R5 | 21.8 | RYO | 9.6 | RYO | 9.7 | RYO | 9.6 |
| R6 | 10.0 | XL | 22.9 | V1 | 52.2 | V1 | 51.1 |
| R7 | 9.6 | HF | 40.4 | V2 | 25.9 | V2 | 26.8 |
| R8 | 13.6 | HKO | 15.2 | V3 | 36.7 | V3 | 38.4 |
| R9 | 25.6 | HKG | 8.3 | V4 | 39.3 | R14 | 18.7 |
| R10 | 18.0 | BO | 16.1 | V5 | 29.5 | R15 | 15.0 |
| R11 | 20.3 | JGS | 15.3 | V6 | 35.2 | R16 | 12.8 |
| R12 | 22.4 | ULD | 15.0 | V7 | 34.7 | R17 | 19.5 |
| R13 | 12.6 | DOK | 14.7 | V8 | 41.6 | R18 | 15.9 |







**Figure 5.** Distribution of self-sensitivity at each observation site for (a) EXP1, (b) EXP2, (c) EXP3, and (d) EXP4. The overlapping observation sites at close locations are distinguished by different sizes of circles.




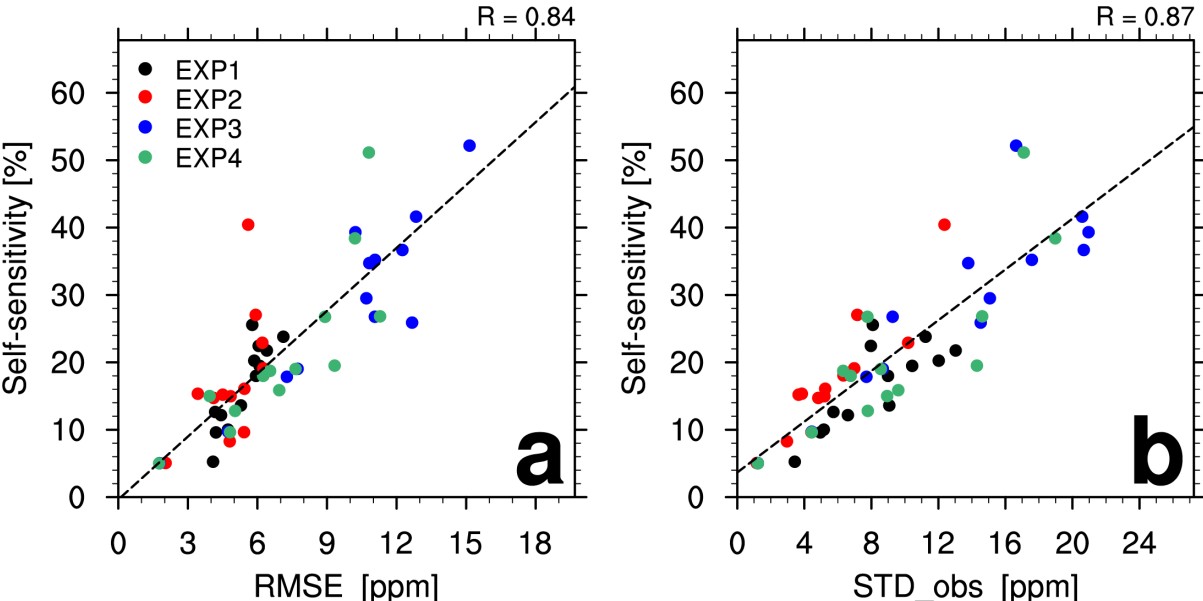

**Figure 6. Scatter plot (a) between RMSE of the analysis [ppm] and self-sensitivity [%] and (b) between standard deviation of**
**hourly surface CO₂ observations [ppm] and self-sensitivity [%] at each observation site for each experiment. The gray dashed line**
**represents the regression line.**

### 3.3. Impact of surface CO₂ observations on forecast

### 3.3.1. Nonlinear forecast error reduction

The reduction of the forecast error of $CO_2$ concentrations when assimilating surface $CO_2$ observations was investigated. Figure 7 shows $e_a$ (Eq. (9)), $e_b$ (Eq. (10)), and NER (Eq. (11)) for the $CO_2$ concentration in each experiment, and the values are presented in Table 5. In all experiments and forecast times, $e_a$ was smaller than $e_b$, which implied that the forecast error of the $CO_2$ concentration was reduced with DA of surface $CO_2$ observations (Figs. 7a and b). The average $e_b$ ($e_a$) for all forecast times was 184.60 (154.04) × 10⁴, 192.64 (159.61) × 10⁴, 191.16 (159.83) × 10⁴, and 188.27 (154.54) × 10⁴ ppm² for EXP1, EXP2, EXP3, and EXP4, respectively (Table 5). The forecast error was reduced by an average of 17.0% by assimilating the surface $CO_2$ observations. Although slight differences existed depending on the forecast time and experiment, both $e_a$ and $e_b$ were greater in EXP2 and EXP3 than those in EXP1 and EXP4. The surface $CO_2$ observation sites in EXP2 and EXP3 were slightly concentrated in certain regions within the domain, whereas those in EXP1 and EXP4 were relatively evenly distributed within the domain by using the Random selection strategy (Fig. 3). Assimilating evenly distributed surface $CO_2$ observations reduced forecast errors in EXP1 and EXP4. The average $e_b$ ($e_a$) for all experiments was 37.68 (8.20) × 10⁴, 92.95 (67.80) × 10⁴, 295.62 (252.79) × 10⁴, and 330.42 (299.23) × 10⁴ ppm² for 6 h, 12 h, 18 h, 24 h forecasts, respectively (Table 5). The forecast errors with and without DA in all experiments increased as the forecast time increased from 6 to 24 h (Figs. 7a and b).





The NER was negative for all forecast times in all experiments (Fig. 7c), which implies that the DA of the surface $CO_2$
concentration observations reduced the errors of the $CO_2$ concentration forecasts. The average NER for all the experiments
was -29.48 × 10$^4$, -25.15 × 10$^4$, -42.83 × 10$^4$, and -31.20 × 10$^4$ ppm$^2$ for 6 h, 12 h, 18 h, and 24 h forecasts, respectively. The
NER increased as the forecast time increased because the forecast errors with and without DA increased. The NER was the
greatest for the 18 h forecast (Table 5), which suggests that the impact of assimilated surface $CO_2$ observations on reducing
the $CO_2$ concentration error was the greatest for 18 h forecast. When the forecast time becomes longer than 18 h, the
nonlinear process within the model may have a greater impact than the improvement of the initial condition accuracy with
the DA, on reducing the forecast error (Kim et al., 2013).

In EXP1 (EXP4), the forecast error with DA decreased by 16.6% (17.9%) compared with that without DA. The observation
impact on forecast error reduction was greater in EXP4 than in EXP1 (Table 5). This was partly due to the greater $e_b$ in
EXP4 than in EXP1, implying more room for reduction in EXP4 than in EXP1. The use of both Variability and Random
strategies in selecting the surface $CO_2$ observation sites in EXP4, compared with only Random strategy used in EXP1, may
be another reason. Selecting observation sites in regions with greater variability in surface $CO_2$ concentrations would have a
greater impact on reducing $CO_2$ forecast errors.

In EXP2 (EXP3), with non-uniform observation sites, the forecast error with DA decreased by 17.2% (16.4%) compared to
that without DA. The observation impact on forecast error reduction was greater in EXP2 than in EXP3 (Table 5). The
impact of the surface $CO_2$ observations added in EXP3 was large in the analysis (Fig. 5c), whereas it was small in forecast
reduction.

**Table 5. Average forecast (fcst) errors without DA ($e_b$), forecast errors with DA ($e_a$), and NER, the difference between $e_b$ and $e_a$ for each experiment. The unit is × 10$^4$ ppm$^2$.**

| Forecast error | EXP1 | | | | EXP2 | | | | EXP3 | | | | EXP4 | | | |
|---|---|---|---|---|---|---|---|---|---|---|---|---|---|---|---|---|
| | 6 h fcst | 12 h fcst | 18 h fcst | 24 h fcst | 6 h fcst | 12 h fcst | 18 h fcst | 24 h fcst | 6 h fcst | 12 h fcst | 18 h fcst | 24 h fcst | 6 h fcst | 12 h fcst | 18 h fcst | 24 h fcst |
| $e_b$ | 37.45 | 91.01 | 288.52 | 321.42 | 38.35 | 95.69 | 302.53 | 333.97 | 37.03 | 91.05 | 302.68 | 333.86 | 37.89 | 94.03 | 288.73 | 332.42 |
| $e_a$ | 8.06 | 66.89 | 247.39 | 293.83 | 8.46 | 70.49 | 257.66 | 301.81 | 8.17 | 66.07 | 260.74 | 304.32 | 8.11 | 67.73 | 245.36 | 296.94 |
| $e_a - e_b$; NER | -29.39 | -24.12 | -41.13 | -27.59 | -29.89 | -25.2 | -44.87 | -32.17 | -28.86 | -24.98 | -41.95 | -29.54 | -29.78 | -26.30 | -43.37 | -35.48 |




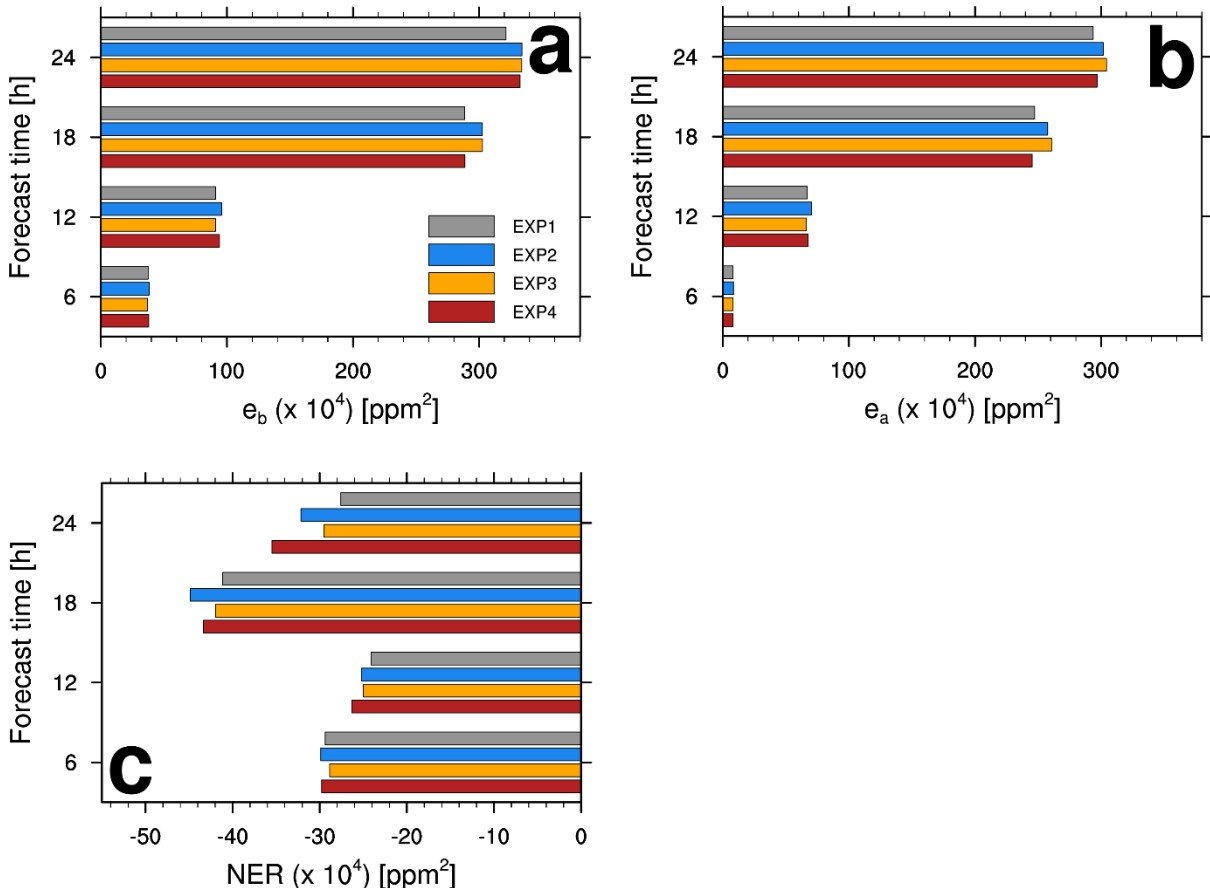

**Figure 7.** Average forecast error (a) with DA, (b) without DA, and (c) the difference between forecast error with DA and without DA (NER) of EXP1 (gray), EXP2 (blue), EXP3 (orange), and EXP4 (red).

### 3.3.2. Linear forecast error reduction

LER (i.e., EFSO) is a linear approximation of NER. The impact of individual observations at each observation site can be evaluated using EFSO. As the nonlinearity in a high-resolution model may increase as the forecast time increases (Gasperoni et al., 2024), the validity of the linear assumption was checked by comparing the NER and EFSO for 6, 12, 18, and 24 h forecasts.

Figure 8 shows the time series of NER and LER for each forecast time in each experiment. The trends of the time series of NER and LER need to be similar when the linear assumption holds appropriately. The time series of the LER and NER were similar for the 6 h and 12 h forecasts (Figs. 8a, e, i, m, b, f, j, and n), whereas the LER fluctuated greatly compared to the NER for the 18 h and 24 h forecasts (Figs. 8c, g, k, o, d, h, l, and p). The linear assumption held appropriately for forecast times up to 12 h.



Figure 9 shows the distributions of the absolute difference between the NER and LER averaged for all experiments for each forecast time. Similar to Fig. 8, the differences between NER and LER for the 6 h and 12 h forecasts were relatively small

compared to those for the 18 h and 24 h forecasts, which showed large differences in inland China. The differences between the NER and LER in the Korean Peninsula and Japan increased for forecast times of 18 and 24 h. Therefore, the validity of the linear assumption held for forecast times up to 12 h.

In a high-resolution convective scale regional model (i.e., WRF) and the GSI-based EnKF DA, Gasperoni et al. (2024) showed that the pattern correlation between NER and EFSO decreased as the forecast time increased owing to the

nonlinearity of the forecasts. Similarly, the linear assumption holds appropriately for forecast times up to 12 h because of the increased nonlinearity of the forecasts; thus, the EFSO was analyzed only for the 6 h and 12 h forecasts.

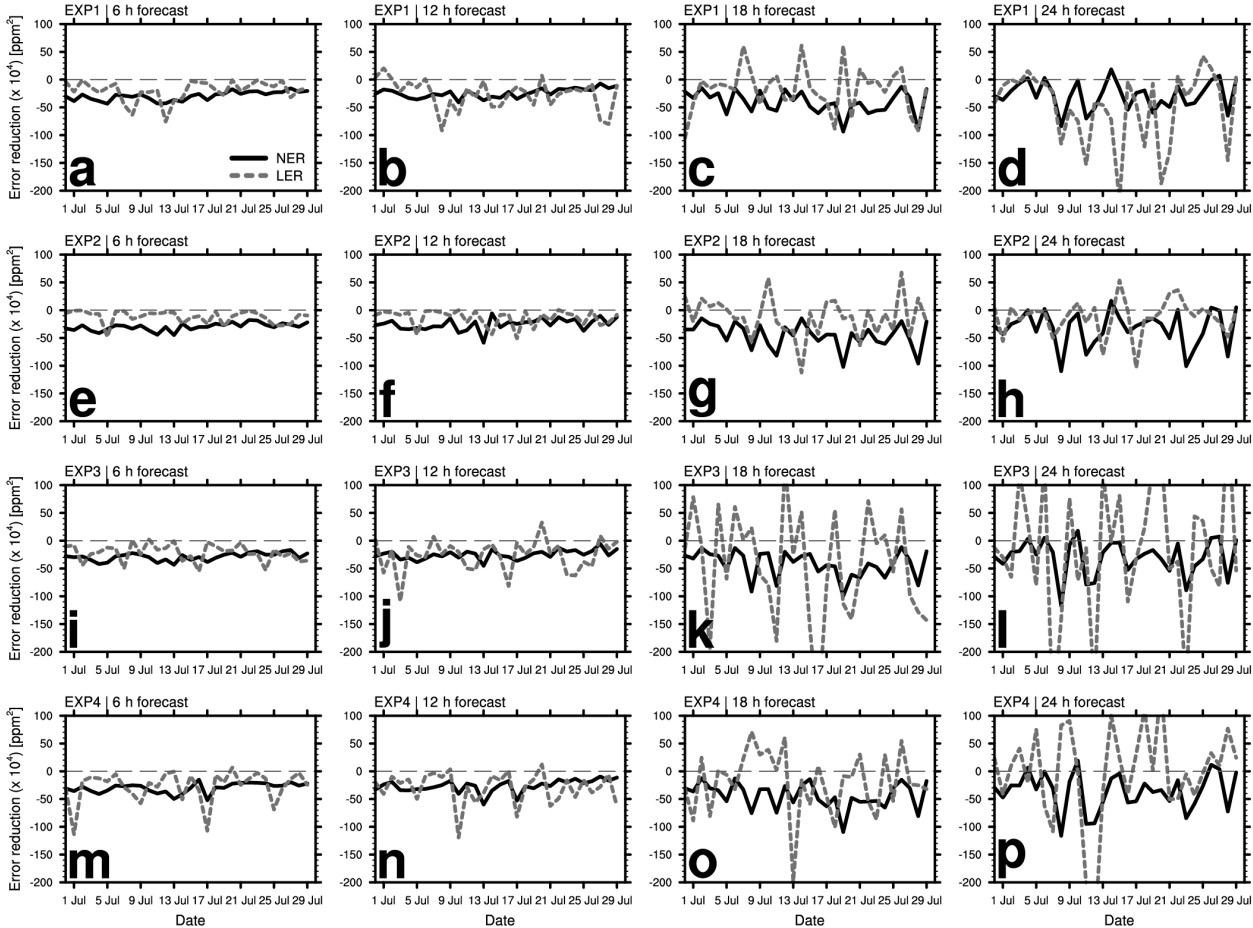

**Figure 8. Time series of NER (black solid) and LER (gray dashed) of (a) EXP1, (e) EXP2, (i) EXP3, (m) EXP4 for 6 h forecast; (b)**
**EXP1, (f) EXP2, (j) EXP3, (n) EXP4 for 12 h forecast; (c) EXP1, (g) EXP2, (k) EXP3, (o) EXP4 for 18 h forecast; and (d) EXP1, (h) EXP2, (l) EXP3, (p) EXP4 for 24 h forecast.**



**Figure 9. Distribution of average difference fields of NER and LER for (a) 6 h forecast, (b) 12 h forecast, (c) 18 h forecast, and (d) 24 h forecast. Each subplot is average distribution of EXP1, EXP2, EXP3, and EXP4.**

### 3.3.3. Ensemble forecast sensitivity to observation

Figures 10 and 11 show the EFSO impacts for the 6 h and 12 h forecasts, respectively. The EFSO impact values for each observation site are listed in Table 6. In all experiments, the EFSO impacts of the surface $CO_2$ observation sites were generally greater for sites with high self-sensitivity.





In EXP1, based on the random selection of observation sites, the EFSO impacts for the 6 and 12 h forecasts at Sites R2, R6, and R7, which were located in the northwestern region of the domain with low $CO_2$ variability, were relatively small (Figs. 10a and 11a), corresponding to the small self-sensitivity shown in Fig. 5a. In contrast, the EFSO impacts in the inland regions of China (i.e., Sites R1, R5, R9, and R13) were greater than those in other regions, which implies that the DA of the surface $CO_2$ observations in the inland regions of China had a greater impact on the error reduction of $CO_2$ concentration forecasts.

The EFSO impacts in EXP2 with real surface $CO_2$ observation sites were relatively small at most observation sites, indicating that the impact of $CO_2$ concentration observations on reducing forecast errors was relatively small when only existing surface $CO_2$ observations were assimilated. The EFSO impacts at Hefei (HF) and Xinglong (XL) located in China were $-6.17 \times 10^4$ and $-3.60 \times 10^4$ $ppm^2$ for 6 h forecast, respectively, and $-5.46 \times 10^4$ and $-4.96 \times 10^4$ $ppm^2$ for 12 h forecast, respectively. The EFSO impacts of HF and XL sites were greater than those of the other sites, implying that surface $CO_2$ observations in inland China, with few observation sites, are essential for reducing $CO_2$ concentration forecast errors in East Asia.

Unlike the large analysis sensitivities, the EFSO impact of EXP3, which added eight surface $CO_2$ observation sites with high variability in $CO_2$ concentrations, was not large. According to Casaretto et al. (2023), the impact of individual observations on reducing forecast errors decreases when the observation density is high in a specific region. Because the surface $CO_2$ observation sites in EXP3 were concentrated in a specific region, the impact of individual observations on reducing forecast errors may not be large.

The EFSO impact of EXP4 was greater at observation sites with greater $CO_2$ concentration variability (i.e., Sites V1, V2, and V3) than at other sites. This implies that the impact on reducing forecast errors was greater when the observation sites were located evenly over the entire domain considering the variability of $CO_2$ concentrations, rather than simply randomly locating observation sites throughout the entire domain.

When analyzing the correlation between the analysis sensitivity and the hourly variability of $CO_2$ concentrations (Fig. 6b), the analysis sensitivity was high at observation sites with high $CO_2$ variability. There was a distinct diurnal variation in atmospheric $CO_2$ concentrations, which were lower during the day and higher at night, owing to the respiration and photosynthetic activities of vegetation. Therefore, the vegetation type of the region within the localization area in the EAKF may be related to the impact of forecast error reduction. Figure 12 shows the eight vegetation types defined by the VPRM used to simulate biogenic $CO_2$ in this study. The representative vegetation types in the domain were *Trees and crops* type; *Grasses* type; and *Barren, urban, built-up, etc.* type. Among the surface $CO_2$ observation sites assimilated in each experiment, the main vegetation types within each localization region were as follows: *Trees and crops* type was dominant in Sites R1, R5, and R9 in EXP1; Sites HF and XL in EXP2; Sites V1, V3, V4, V5, V6, V7, and V8 in EXP3; and Sites V1, V3, R14, and R17 in EXP4. *Grasses* type was dominant in Sites R2, R6, and R8 in EXP1, Site V2 in EXP3, and Sites V2 and R16 in EXP4. *Barren, urban, built-up, etc.* type was dominant at other sites.



In both the 6 h and 12 h forecasts, the EFSO impacts of surface $CO_2$ observation sites dominated by *Trees and crops* type were high, and the EFSO impacts of surface $CO_2$ observation sites dominated by *Grasses* and *Barren, urban, built-up, etc.* types were small. Vegetation activities in *Trees and crops* type were higher than those in *Grasses* and *Barren, urban, built-up, etc.* types, thus the magnitude of daily variabilities of $CO_2$ concentrations was also greater in *Trees and crops* type than in *Grasses* and *Barren, urban, built-up, etc.* types. Therefore, compared to surface $CO_2$ observation sites in other vegetation

types, the impact of observations located in regions with active vegetation activities, such as the *Trees and crops* type, was greater in reducing the 6 h and 12 h forecast errors. In other words, the improvement of the initial condition of $CO_2$ concentrations by DA of surface $CO_2$ concentrations in *Trees and crops* type greatly reduced the 6 h and 12 h forecast errors. Figure 13 shows the fraction of beneficial observations for 6 h and 12 h forecasts in each experiment. The fraction of beneficial observations is the ratio of the number of observations that reduced the forecast error to the total number of

observations assimilated at each observation site. In general, the fraction of beneficial observations of meteorological variables in FSO and EFSO studies is between 55% and 65% (Kim et al., 2017b; Lien et al., 2018; Necker et al., 2018; Kim and Kim, 2021; Casaretto et al., 2023). In some cases, the fraction of beneficial observations exceeds 70% (Jung et al., 2013). The average fractions of beneficial observations for the four experiments were 68.9% and 66.3% for the 6 h and 12h forecast, respectively. On average, more than half of the observations contributed to reducing the forecast errors. In all experiments,

the fraction of beneficial observations for the 6 h forecast was greater than that for the 12 h forecast. For the 12 h forecast, the fractions of beneficial observations for Sites R10 and R14 selected randomly and Sites V1 and V4 selected by "Variability" were less than 50%. The EFSO impact of Site R10 in EXP1 was $0.27 \times 10^4$ $ppm^2$ (greater than 0) for the 12 h forecast, which implies that observations at Site R10 contributed to increasing the 12 h forecast error. In contrast, for the 12 h forecast, the EFSO impacts were $-3.40 \times 10^4$ $ppm^2$ at Site V1 of EXP3, $-1.91 \times 10^4$ $ppm^2$ at Site V4 of EXP3, $-4.66 \times 10^4$

$ppm^2$ at Site V1 of EXP4, and $-8.49 \times 10^4$ $ppm^2$ at Site R14 in EXP4, contributing to a reduction in the forecast error. More than half of the observations at Sites A1, A4, and R14 did not contribute to reducing forecast errors for the 12 h forecast, whereas the average EFSO impact of Sites A1, A4, and R14 negatively contributed to reducing the forecast error. Therefore, on average, more than half of the assimilated surface $CO_2$ observations in the four experiments contributed to the reduction of forecast errors.




**Table 6. Average EFSO impact for each experiment. The unit is $\times 10^4$ ppm$^2$.**

| EXP1 | | | EXP2 | | | EXP3 | | | EXP4 | | |
|---|---|---|---|---|---|---|---|---|---|---|---|
| Site ID | 6 h fcst | 12 h fcst | Site ID | 6 h fcst | 12 h fcst | Site ID | 6 h fcst | 12 h fcst | Site ID | 6 h fcst | 12 h fcst |
| R1 | -6.33 | -7.87 | YON | 0.00 | 0.00 | YON | -0.03 | -0.01 | YON | 0.00 | 0.00 |
| R2 | 0.02 | -0.30 | AMY | -0.64 | -1.94 | AMY | -1.82 | -3.96 | AMY | -1.18 | -1.12 |
| R3 | -3.00 | -2.90 | DDR | -0.40 | -0.42 | DDR | -0.28 | -0.29 | DDR | -0.52 | -0.28 |
| R4 | -0.41 | -0.26 | KIS | -0.47 | -0.71 | KIS | -0.05 | -0.49 | KIS | 0.14 | -0.12 |
| R5 | -2.61 | -5.46 | RYO | -0.48 | -0.22 | RYO | -0.47 | -0.29 | RYO | -0.43 | -0.22 |
| R6 | -0.62 | -0.84 | XL | -3.60 | -4.96 | V1 | -4.73 | -3.40 | V1 | -7.62 | -4.66 |
| R7 | -0.21 | -0.79 | HF | -6.17 | -5.46 | V2 | -3.91 | -4.44 | V2 | -2.44 | -8.63 |
| R8 | -2.16 | -2.69 | HKO | -1.13 | -1.97 | V3 | -3.70 | -5.69 | V3 | -7.34 | -6.65 |
| R9 | -4.24 | -3.71 | HKG | -1.55 | -2.02 | V4 | -3.00 | -1.91 | R14 | -2.22 | -8.49 |
| R10 | -0.32 | 0.27 | BO | -1.06 | -1.32 | V5 | -4.33 | -5.32 | R15 | -1.81 | -1.62 |
| R11 | -2.69 | -2.51 | JGS | -0.44 | -0.27 | V6 | -1.96 | -3.04 | R16 | -2.25 | -2.85 |
| R12 | -1.82 | -3.98 | ULD | -0.89 | -1.25 | V7 | -1.97 | -2.58 | R17 | -12.24 | -8.80 |
| R13 | -0.38 | -0.84 | DOK | -0.35 | -0.24 | V8 | -3.62 | -4.77 | R18 | -2.97 | -3.85 |






**Figure 10.** Distribution of EFSO impact on the 6 h forecast at each observation site of (a) EXP1, (b) EXP2, (c) EXP3, and (d) EXP4. Overlapping observation sites at close locations are distinguished by different sizes of circles.



**Figure 11. Distribution of EFSO impact on the 12 h forecast at each observation site of (a) EXP1, (b) EXP2, (c) EXP3, and (d) EXP4. Overlapping observation sites at close locations are distinguished by different sizes of circles.**



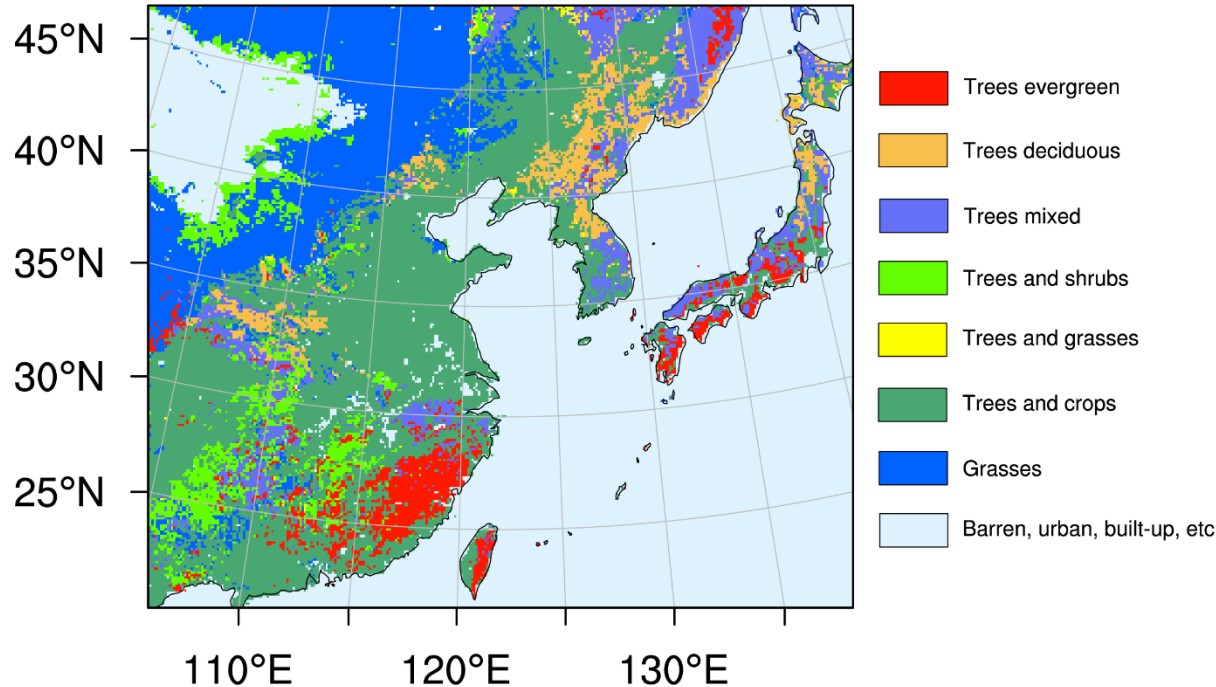

**Figure 12. Distribution of vegetation types used in VPRM.**





**Figure 13. Fraction of beneficial observations of (a) EXP1, (c) EXP2, (e) EXP3, and (g) EXP4 for 6 h forecast and that of (b) EXP1, (d) EXP2, (f) EXP3, and (h) EXP4 for 12 h forecast. The gray dashed line represents a ratio of 50%.**



## 4. Summary and conclusions

In this study, the impact of surface $CO_2$ concentration observations on analysis and forecast error reduction was investigated when optimizing surface $CO_2$ concentrations in East Asia. WRF-Chem, a high-resolution atmospheric chemistry model, was used to simulate the $CO_2$ concentrations, and the EAKF DA of the modified DART was used to assimilate the surface $CO_2$ concentration observations. By assimilating surface $CO_2$ concentration observations using the DA-forecast system that combines WRF-Chem and DART, the impacts of each assimilated surface $CO_2$ observation on the analysis of $CO_2$ concentration and on the forecast error reduction of $CO_2$ concentration were investigated. Pseudo surface $CO_2$ observations extracted using OSSE were used to calculate the observation impact depending on the observation site locations. Four experiments (EXP1, EXP2, EXP3, and EXP4) were conducted that assimilated the pseudo surface $CO_2$ concentration observations at four surface $CO_2$ observation networks. The experiment was conducted during July 2019.

A rank histogram was analyzed to investigate whether the four ensemble DA-forecast experiments were stable during the experimental period. The rank histogram was flat and the adjusted missing rate of the rank histogram was less than 10% in all experiments. Thus, the ensemble DA-forecast system performed appropriately in all the experiments.

Self-sensitivity, which is the diagonal component of the influence matrix, was calculated to diagnose the impact of the assimilated surface $CO_2$ concentration observations on the analysis after DA. The average self-sensitivities were 16.5%, 17.5%, 28.7%, and 21.3% in EXP1, EXP2, EXP3, and EXP4, respectively. The average self-sensitivity was the greatest in EXP3, in which the observation sites were located in regions with high variability in surface $CO_2$ concentrations. Both the RMSE of the analysis at each observation site and the hourly variability of the pseudo surface $CO_2$ observations were highly correlated with the self-sensitivity. In other words, the analysis sensitivity increased as the RMSE of the analysis at the observation site and the hourly variability in $CO_2$ concentrations increased when the surface $CO_2$ observations were assimilated with a cycling interval of 6 h.

To investigate the impact of assimilated surface $CO_2$ concentration observations on reducing the 6, 12, 18, and 24 h forecast errors, forecast errors with and without DA and NER were calculated. In all experiments and at all forecast times, the forecast error with DA was smaller than that without DA; thus, the error of the $CO_2$ concentration forecast decreased after the DA of the surface $CO_2$ concentration observations. Forecast errors with and without DA were smaller in EXP1 and EXP4 with uniformly distributed surface $CO_2$ observation sites, whereas they were greater in EXP2 and EXP3 with unevenly distributed surface $CO_2$ observation sites. The forecast errors with DA were reduced by 16.6%, 17.2%, 16.4%, and 17.9% in EXP1, EXP2, EXP3, and EXP4, respectively, compared with the forecast error without DA. Among EXP1 and EXP4 with smaller forecast errors, the observation impact on reducing forecast errors was greater in EXP4. The forecast error and observation impact were small for experiments with relatively evenly distributed observation sites, whereas they were large for experiments with concentrated observation sites in a specific region.

The LER (i.e., EFSO) was compared with the NER to verify the suitability of LER. From the 18 h forecast, the difference between the NER and LER increased, reducing the reliability of the EFSO. This is because the nonlinearity of the model



increased as the forecast time increased beyond 18 h. The impact of surface $CO_2$ observations on reducing the 6 h and 12 h forecast errors was generally greater at observation sites with greater analysis sensitivity. The EFSO impact was greater in

EXP4 than in EXP1, EXP2, and EXP3. This implies that surface $CO_2$ observations can reduce forecast errors the most when surface $CO_2$ observation sites cover the entire domain and are located considering the variability characteristics of the $CO_2$ concentrations. Surface $CO_2$ observation sites located in regions with more active vegetation types (i.e., *Trees and crops* type) within the localization region greatly reduced forecast errors. At surface $CO_2$ observation sites with greater daily fluctuations in $CO_2$ concentration due to vegetation activity during the day and night, the improvement in the initial condition accuracy

by DA greatly reduced the forecast errors.

The average fractions of beneficial observations were 68.9% and 66.3% for the 6 and 12 h forecast, respectively, indicating that more than half of the surface $CO_2$ observations contributed to reducing forecast errors.

In the future, the impact of various $CO_2$ observation types could be evaluated by assimilating aircraft and satellite $CO_2$ observations, in addition to surface $CO_2$ observations. In addition, the impact of assimilated $CO_2$ observations on the

estimation of surface carbon flux can be evaluated. The results of this study and future studies will be useful for monitoring and estimating atmospheric $CO_2$ concentrations, optimizing surface $CO_2$ fluxes, and designing atmospheric $CO_2$ observation networks.

**Code/data availability**

The Weather Research and Forecasting model coupled with Chemistry (WRF-Chem) code are available at https://doi.org/10.5065/D6MK6B4K (https://github.com/wrf-model/WRF/releases) and the Data Assimilation Research Testbed (DART) code are available at http://doi.org/10.5065/D6WQ0202 (https://github.com/NCAR/DART). The CarbonTracker version 2022 (CT2022) data are available at https://dx.doi.org/10.25925/z1gj-3254 and the Copernicus Atmosphere Monitoring Service (CAMS) data are available at https://ads.atmosphere.copernicus.eu/datasets/cams-global-

ghg-reanalysis-egg4.

**Author contribution**

H. M. Kim proposed the main scientific ideas and M.-G. Seo contributed the supplementary ideas during the process. M.-G. Seo and H. M. Kim developed the $CO_2$ modeling and data assimilation system and calculated the observation impact. All authors analyzed the simulation results, wrote the manuscript, and reviewed the manuscript.





**Competing interests**

The authors declare that they have no conflict of interest.

**Acknowledgements**

This study was supported by the Yonsei Signature Research Cluster Program of 2024 (2024-22-0162). Simulations were primarily conducted using the supercomputer system supported by the National Center for Meteorological Supercomputer of

the Korea Meteorological Administration and Korea Research Environment Open NETwork (KREONET) supported by the Korea Institute of Science and Technology Information.



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
