# Peer review of "Ensemble-based observation impact of surface CO2 concentration observations on analysis and forecast of atmospheric CO2 concentrations over East Asia"

_EGUsphere, 2025_

## Referee Comment (RC1)

**General comments**

The authors present observation system simulation experiments of regional $CO_2$ concentration data assimilation and analyze the impact of different station networks on the forecast quality. The work provides interesting insights for high-resolution, regional $CO_2$ forecasting based on in-situ observations. These insights deserve publication.

However, the discussion of the results should be extended with the aim of understanding the implications for other setups and possibly for forecasts with real observations. I do not expect a long discussion of OSSE in general, and I do see that the authors discuss and evaluate some important aspects of their setup, e.g., using the rank histograms. But I miss an overall discussion which aspects of the setup the authors deem crucial for the interpretability and transferability of the results. Just to provide an example, the construction of the true state in the OSSE might be a relevant aspect (see my comments concerning line 233). Ideally, I would expect a concise discussion of these aspects in the conclusions section.

The setup is in general well structured and well explained, aside from a few remaining questions listed in the specific comments. The presentation of the results is well understandable, but parts of the results could be presented more concisely (see my comment concerning lines 468–484).

**Specific comments**

**line 139, beginning of section 2.3.2** In section 2.3.1, the authors define the self-sensitivity, which is defined in observation space. In section 2.3.2, a similar vector notation as for the observation space is used for the state space. A brief comment on the structure of the state vectors ($x^f$ and $x_t$) could guide the reader to immediately see this difference.

**line 155, Eq. (13)** The definition of $\delta y_0$ is missing. I assume that $\delta y_0 = y - H\bar{x}_b$.

**line 233, emissions in OSSE** The choice of emissions for the OSSE true state and for the DA experiments is mentioned in sections 2.4.1 and 2.4.2. This choice is important to obtain meaningful results, as indicated by the authors when mentioning the identical twin problem (line 234). My impression is that the comparison of the emissions chosen for generating the true state and those use for the DA experiment deserves more attention.

The importance of the choice of emissions is illustrated by the following interpretation of the results: The simulation of the true state and the four DA experiments used VPRM and will therefore show very similar (or identical?) biogenic fluxes. The authors find that observation sites in regions with strong biogenic fluxes greatly reduce forecast errors (line 543). Is this because the fluxes in these regions were close to the true fluxes by construction? Or do the authors expect a similar improvement in an experiment with real observations?

I suggest to state explicitly how the emissions in the true state and in the DA experiments differ. If the authors agree on the relevance of this choice of emissions for the results, this aspect should be mentioned when presenting or discussing the results. (see also my general comments above)

**line 191** The authors write: "The $CO_2$ variability of the true state in this study was included in the $CO_2$ variabilities of CT2022 and the Copernicus Atmosphere Monitoring Service (CAMS), ..." It is not clear to me how CAMS is used and how the variabilities of CT2022 are considered to create a reasonable deviation of the DA experiments from the true state.

**line 256** The authors list the meteorological and chemical initial and lateral boundary conditions for the experiments without distinguishing between the single forecast run for the true state and the ensemble forecast for the DA experiments. Were the lateral boundary conditions identical

identical for all model runs? If yes, do the authors expect an underestimated deviation from the true state at the lateral boundaries that could influences the results?

How did the authors make sure that the spread of the meteorological ensemble remains approximately constant? Was an initial condition update cycling or similar technique used?

**line 354 (minor comment)** The authors provide many values with the unit $10^4\,\mathrm{ppm}^2$, to be interpreted as the sum of squared errors in all grid cells. A (very) brief guidance on how to interpret these numbers could possibly help the reader in section 3.3.1 and in Table 5.

**lines 354 and 361 (minor comment)** The authors provide many numbers and even lists of numbers in the text. I do not want to criticise what is a matter of style, but instead of reading lists of 8 numbers as in lines 354 and 361, I personally prefer the structure of a table. The paper contains many tables already and some of the numbers provided in the text could be included in the existing tables, e.g., by including averages in Table 5.

**lines 424 to 484 (minor comment)** Section 3.3.3 mainly discusses three separate aspects: The correlation of EFSO with self-sensitivity and $CO_2$ concentration variability; the impact of vegetation types around the observation sites; and the fraction of beneficial observations. For the reader it might be easier to follow if section 3.3.3 is split into two or even three parts (e.g. start a new section 3.3.4 on line 468 for the fraction of beneficial observations).

**lines 468 to 484 (minor comment)** This paragraph could be written more concisely. For example, the following sentences seem redundant: "The average fractions of beneficial observations … were 68.9% and 66.3% … On average, more than half of the observations contributed to reducing the forecast errors. … Therefore, on average, more than half of the assimilated surface $CO_2$ observations in the four experiments contributed to the reduction of forecast errors."

In general, the presentation of the results is clear and understandable. But as the example shows, it could be more concise in a few paragraphs.

**Technical corrections**

**line 104, Eq. (4)** I do not understand how Eq. (4) leads to $A^\top$ in Eq. (3) (instead of just $A$).

**Fig. 1** The figure is very illustrative, but on the horizontal axis, the label "t h" seems odd to me since $t$ is usually considered a value including a unit.

**line 284** I was confused by the word "multiple", which is meant here in the sense of "too many".

**line 381** "forecast reduction" should probably be "forecast error reduction"

**lines 481 f.** Sites "A1, A4" should probably be "V1, V4"

---

## Author Comment (AC1)

**Egusphere-2025-2367 (Editor: Christoph Gerbig)**

**Response to Reviewer 1**

The authors thank the reviewer 1 for a thoughtful review of the manuscript. Considering the reviewer's points, we have made the necessary changes. The responses for the reviewer's specific comments are as follows.

**General comments:**

*The authors present observation system simulation experiments of regional $CO_2$ concentration data assimilation and analyze the impact of different station networks on the forecast quality. The work provides interesting insights for high-resolution, regional $CO_2$ forecasting based on in-situ observations. These insights deserve publication.*
*However, the discussion of the results should be extended with the aim of understanding the implications for other setups and possibly for forecasts with real observations. I do not expect a long discussion of OSSE in general, and I do see that the authors discuss and evaluate some important aspects of their setup, e.g., using the rank histograms. But I miss an overall discussion which aspects of the setup the authors deem crucial for the interpretability and transferability of the results. Just to provide an example, the construction of the true state in the OSSE might be a relevant aspect (see my comments concerning line 233). Ideally, I would expect a concise discussion of these aspects in the conclusions section.*
*The setup is in general well-structured and well explained, aside from a few remaining questions listed in the specific comments. The presentation of the results is well understandable, but parts of the results could be presented more concisely (see my comment concerning lines 468−484).*

> **Authors' response**: The authors thank the reviewer 1 for a thoughtful review of the manuscript. As the reviewer denoted, the experimental design of the observing system simulation experiment (OSSE) is important for the interpretability and transferability of our results and possibly for forecasts with real observations. To this end, realistic differences between the true state and the data assimilation (DA) experiments were ensured by using different $CO_2$ emission input data.
>
> For the anthropogenic and oceanic $CO_2$, different emission datasets were used for the true state and the DA experiments. For the biogenic $CO_2$, while the Vegetation Photosynthesis and Respiration Model (VPRM) was used in true state and DA experiments, different parameter tables in VPRM were used to calculate biogenic fluxes, which creates considerable differences. We expect that this experimental design allows the pseudo $CO_2$

observations extracted from the true state to operate similarly to real observations. More detailed discussions are shown in the authors' response to specific comments 3.

Regarding the suggestion to present parts of the results section more concisely, please refer to the authors' response to specific comments 9.

**Specific comments:**

*1. **line 139, beginning of section 2.3.2** In section 2.3.1, the authors define the self-sensitivity, which is defined in observation space. In section 2.3.2, a similar vector notation as for the observation space is used for the state space. A brief comment on the structure of the state vectors ($x^f$ and $x_t$) could guide the reader to immediately see this difference.*

**Authors' response:** As the reviewer denoted, the self-sensitivity in section 2.3.1 was calculated using state vectors in the observation space (i.e., $\mathbf{HX}_a$: analysis ensemble perturbation in the observation space), whereas the observation impact in section 2.3.2 was calculated using state vectors in model space (i.e., $\mathbf{x}^f$ and $\mathbf{x}_t$). To clarify this difference, we have revised the text as follows. The revised parts are underlined.

Line 188-191 in the revised manuscript: "Forecast error is calculated as follows:

$$e = (\mathbf{x}^f - \mathbf{x}_t)^{\mathbf{T}}\mathbf{C}(\mathbf{x}^f - \mathbf{x}_t), \tag{8}$$

where $\mathbf{x}^f$ is the forecast, $\mathbf{x}_t$ is the true state, and $\mathbf{C}$ is the positive definite matrix that defines the norm. $\underline{\mathbf{x}^f \text{ and } \mathbf{x}_t \text{ are state vectors in the model space.}}$"

*2. **line 155** Eq. (13) The definition of $\delta\mathbf{y}_0$ is missing. I assume that $\delta\mathbf{y}_o = \mathbf{y} - \mathbf{H}\bar{\mathbf{x}}_b$.*

**Authors' response:** We have added the definition of $\delta\mathbf{y}_o$ in the revised manuscript as follows. The added parts are underlined.

Line 203-206 in the revised manuscript: "where $\mathbf{M}$ is the tangent linear model of the nonlinear NWP model, $\mathbf{X}_a$ is the ensemble perturbation of analysis, $\mathbf{X}_a^f$ is the ensemble perturbation of forecast integrated from the analysis, $\mathbf{Y}_a$ is the ensemble perturbation of analysis in observation space, satisfying the relationship of $\mathbf{Y}_a \approx \mathbf{HX}_a$ (Note Eq. (6)). $\underline{\delta\mathbf{y}_o \text{ is the observational increment compare to the first guess, satisfying the relationship}}$ $\underline{\text{of } \delta\mathbf{y}_o = \mathbf{y} - \mathbf{H}\bar{\mathbf{x}}_b.}$"

*3. **line 233, emissions in OSSE** The choice of emissions for the OSSE true state and for the DA experiments is mentioned in sections 2.4.1 and 2.4.2. This choice is important to obtain*

*meaningful results, as indicated by the authors when mentioning the identical twin problem (line 234). My impression is that the comparison of the emissions chosen for generating the true state and those use for the DA experiment deserves more attention.*

*The importance of the choice of emissions is illustrated by the following interpretation of the results: The simulation of the true state and the four DA experiments used VPRM and will therefore show very similar (or identical?) biogenic fluxes. The authors find that observation sites in regions with strong biogenic fluxes greatly reduce forecast errors (line 543). Is this because the fluxes in these regions were close to the true fluxes by construction? Or do the authors expect a similar improvement in an experiment with real observations?*

*I suggest to state explicitly how the emissions in the true state and in the DA experiments differ. If the authors agree on the relevance of this choice of emissions for the results, this aspect should be mentioned when presenting or discussing the results. (see also my general comments above)*

**Authors' response:** In this study, to avoid the identical twin problem, different emission data were used for true state and DA experiments (Table_rev1_1). In VPRM simulations of biogenic $CO_2$ in WRF-Chem, empirical parameters $\alpha$, $\beta$, $\gamma$, and $PAR_0$ should be optimized for each land use type in the experimental region (Hilton et al. 2013). Although VPRM module was used to simulate biogenic $CO_2$ fluxes in true state and DA experiments, four parameters ($\alpha$, $\beta$, $\gamma$, and $PAR_0$) in VPRM were different in true state and DA experiments, which provides sufficient differences in the biogenic $CO_2$ concentrations in true state and DA experiments. In this study, based on Seo et al. (2024), which analyzed the effect of VPRM parameters on $CO_2$ simulations over East Asia, the US parameter table and Li parameter table (Li et al. 2020) were used for true state and DA experiments, respectively.

Table_rev1_1. Description of emission data used in each experiment.

| Experiments | Emission input | | | DA |
| --- | --- | --- | --- | --- |
| | Anthropogenic | Biogenic | Oceanic | |
| True state | Average of CT2022 and ODIAC | VPRM * US table | CT2022 | X |
| DA experiments (EXP1, EXP2, EXP3, and EXP4) | ODIAC | VPRM * Li table | JMA ocean map | O |

Figure_rev1_1 shows the distributions of anthropogenic and biogenic $CO_2$ concentrations simulated in true state and DA experiments, along with the spatial distribution of absolute relative differences of DA experiments from the true state. The distribution in DA

experiments is the average distribution of four DA experiments. The average difference in anthropogenic $CO_2$ concentrations between true state and DA experiments is 2.7%, and that of biogenic $CO_2$ concentrations is 2.2%, indicating that the biogenic $CO_2$ concentrations in true state and DA experiments are not identical and differ similar magnitudes as the anthropogenic $CO_2$ concentrations.

Therefore, the substantial reduction in forecast errors at observation sites located in regions with strong biogenic fluxes is not due to the biogenic fluxes being identical between the true state and the DA experiments.

[Figure]

Figure_rev1_1. Distribution of average anthropogenic $CO_2$ concentrations (ppm) in (a) true state and (b) four DA experiments, and average biogenic $CO_2$ concentrations (ppm) in (d) true state and (e) four DA experiments. Absolute relative difference (%) distribution between DA experiments and true state based on true state for (c) anthropogenic and (f) biogenic $CO_2$ concentrations.

In addition, if the biogenic flux is identical in true state and DA experiments, then the regions with large biogenic flux will appear similarly in both true state and DA experiments, and the differences between the two will be small. When data assimilation is done in

regions where the absolute magnitude of the flux is large with small differences in two types of experiments (i.e., true state and DA experiments), it is difficult to significantly reduce the forecast error in those regions, because the forecast error—defined as the difference between the true state and DA experiments—was small. In this study, data assimilation reduced forecast errors greatly in regions with strong biogenic fluxes, which implies that the flux differences in those regions are large and that the modules used to calculate the biogenic fluxes in two experiments (i.e., true state and DA experiments) were different in those regions.

To clarify the differences in the VPRM parameter tables used in each experiment, we have revised the text as follows. The revised parts are underlined.

Line 286-294 in the revised manuscript: "The emission data used in the four experiments differed from those used to simulate the true state. This is to avoid the identical twin problem that can occur in the OSSE by setting the experimental design of the true state and four experiments sufficiently different (Masutani et al., 2010; Shu et al., 2023; Kim et al., 2022). In the four experiments, ODIAC v2020b (Oda and Maksyutov, 2015) was used for anthropogenic emission, the Japan Meteorological Agency (JMA) ocean map (Iida et al., 2021; Takatani et al., 2014) was used for oceanic emission, and VPRM was used for biogenic emission. To ensure sufficient differences between the biogenic $CO_2$ in the true state and in the four experiments, different VPRM parameter tables were used: the US table for the true state and the Li table (Li et al., 2020) for the four experiments. Seo et al. (2024a) showed that biogenic $CO_2$ fluxes in East Asia vary considerably depending on the VPRM parameter tables used, and the parameter values for the US and Li tables are described in detail in Seo et al. (2024a)."

*4. [line 191] The authors write: "The $CO_2$ variability of the true state in this study was included in the $CO_2$ variabilities of CT2022 and the Copernicus Atmosphere Monitoring Service (CAMS), ..." It is not clear to me how CAMS is used and how the variabilities of CT2022 are considered to create a reasonable deviation of the DA experiments from the true state.*

**Authors' response:** In the OSSE framework, the nature run (NR) is assumed to be the true state and should reasonably reflect the variability of real atmospheric states (Yang et al., 2014). Accordingly, the $CO_2$ concentrations in the true state should reflect the variability of real atmospheric $CO_2$ concentrations.

Seo and Kim (2025) employed the same true state (i.e., NR) as in this study and examined whether $CO_2$ concentrations in the NR appropriately represent the variability of real atmospheric $CO_2$ concentrations. For this purpose, the variability of $CO_2$ concentrations in the NR was compared with that in CarbonTracker 2022 (CT2022) and the Copernicus Atmosphere Monitoring Service (CAMS), which are widely used $CO_2$ reanalysis datasets.

The mean and standard deviation of $CO_2$ concentrations in the NR fell within the variability range represented by the standard deviations of CT2022 and CAMS (Figure 2 in Seo and Kim (2025)). This implies that the $CO_2$ concentrations in the NR reasonably reflect the variability of real atmospheric $CO_2$ concentrations.

To clarify, we have revised the text as follows. The revised parts are underlined.

Line 242-246 in the revised manuscript: "In OSSE, the true state must not deviate from the real natural variability (Masutani et al., 2010). The mean and standard deviation of $CO_2$ concentrations in the true state of this study fell within the variability range represented by the standard deviations of CT2022 and the Copernicus Atmosphere Monitoring Service (CAMS), which are widely used $CO_2$ reanalysis datasets (see Fig. 2 in Seo and Kim (2025)). This indicates that the $CO_2$ concentrations in the true state reasonably reflect the variability of real atmospheric $CO_2$ concentrations."

[Figure]

Figure 2 in Seo and Kim (2025). Mean (black line) and standard deviation (black bar) of surface $CO_2$ concentrations in the nature run (NR); mean (red line) and standard deviation (red shading) of surface $CO_2$ concentrations in the CT2022: (a) January 2019 and (c) July 2019. Mean (black line) and standard deviation (black bar) of surface $CO_2$ concentrations in the NR; mean (blue shading) of surface $CO_2$ concentrations in the Copernicus atmosphere monitoring service (CAMS): (b) January 2019 and (d) July 2019.

*5. [line 256] The authors list the meteorological and chemical initial and lateral boundary conditions for the experiments without distinguishing between the single forecast run for the true state and the ensemble forecast for the DA experiments. Were the lateral boundary conditions identical for all model runs? If yes, do the authors expect an underestimated deviation from the true state at the lateral boundaries that could influences the results?*

*How did the authors make sure that the spread of the meteorological ensemble remains approximately constant? Was an initial condition update cycling or similar technique used?*

**Authors' response:** We addressed the three types of the reviewer's questions as follows.

1) Lateral boundary conditions for the experiments

In all experiments (the true state and four DA experiments), final analysis (FNL) was used as the meteorological lateral boundary condition, and CT2022 was used as the chemical lateral boundary condition. The purpose of this study is to investigate the effect of assimilating $CO_2$ observations on analysis and forecast of $CO_2$ concentrations. If the types of lateral boundary conditions are different for the true state and the DA experiments, the effect of assimilating $CO_2$ observations (i.e., observation impact) was conflated with the effect of using different lateral boundary conditions, making it difficult to investigate the pure observation impact on analysis and forecast of $CO_2$ concentrations.

According to Kim and Kim (2021), which analyzed the effect of lateral boundary conditions on the observation impact in a regional model (i.e., WRF), the greater the influence of lateral boundary conditions on the forecast error, the smaller the observation impact. Therefore, in this study, the types (i.e., FNL and CT 2022) of lateral boundary conditions were identical in all experiments to clearly analyze the effect of the assimilated $CO_2$ observations.

Although the types (i.e., FNL and CT 2022) of lateral boundary conditions were identical in all experiments, we did use perturbed lateral boundary conditions for ensemble forecasts in DA experiments, which is different from the single forecast run for true state using a single unperturbed lateral boundary condition. When performing the 20 members of ensemble forecasts, 20 different perturbations in boundary conditions were applied on 20 runs using pert_wrf_bc and update_wrf_bc program from the Data Assimilation Research Testbed (DART) to prevent a decrease in the ensemble spread.

We have revised the text as follows. The revised parts are underlined.

Line 155-157 in the revised manuscript: "Initial perturbations of the chemical variables (i.e., $CO_2$ concentrations) were produced using the method presented in Yumimoto (2013) and Miao (2014). The lateral boundary conditions for the 20 ensemble members were perturbed using the pert_wrf_bc and update_wrf_bc program in DART."

2) The spread of the meteorological ensembles

To appropriately maintain the ensemble spread of both meteorological variables and $CO_2$ concentrations throughout the entire experimental period, four methods were used: (a) initial perturbations, (b) perturbations in boundary conditions, (c) inflation, and (d) multi physics options.

a) Initial perturbations were applied not only to chemical variables (i.e., $CO_2$) but also meteorological variables for the 20 ensemble members. For the meteorological variables, perturbations were randomly selected from 150 perturbation banks generated based on be.dat.cv3 in the Weather Research Forecasting model data assimilation system (WRFDA).

b) The lateral boundary conditions for 20 ensemble members were perturbed using the pert_wrf_bc and update_wrf_bc program in DART.

c) Both prior and posterior inflation were applied to meteorological variables and $CO_2$ concentrations.

d) The multi physics options, which were used to maintain the ensemble spread, influenced the spread of the meteorological variables in each ensemble member by applying different combinations of microphysics, cumulus, and planetary boundary layer schemes.

Above information was already described in Section 2.2. In addition, we calculated the ensemble standard deviation for meteorological variables. The ensemble standard deviation for 10 m U, 10 m V, and 2 m temperature remained stable throughout the experimental period (Figure_rev1_2).

To clarify that the spread of the meteorological variables was consistently maintained throughout the experimental period, we have added a text to the manuscript. The added parts are underlined.

Line 161-166 in the revised manuscript: "The multiplicative inflation of the RTPS was set to 1.0, which was found to be the most appropriate in sensitivity tests. To maintain ensemble spread, multi physics options were also applied for microphysics, cumulus, and planetary boundary layer schemes when conducting ensemble forecasts. The physical parameterization schemes used for the ensemble forecasts are listed in Table 1. Through initial perturbations, boundary condition perturbations, inflation, and multi physics schemes, the ensemble spreads of $CO_2$ concentrations and meteorological variables were properly maintained throughout the experimental period."

3) Initial condition update cycling

Data assimilation cycles updating initial conditions were used in this study. A cycling run was performed at 6-hour intervals, where the analysis generated from the data assimilation was used as the initial condition for the subsequent forecast cycle. To clarify, we have revised the text as follows. The revised parts are underlined.

Line 146-150 in the revised manuscript: "The experimental settings for assimilating the surface $CO_2$ concentration using EAKF were as follows: As diurnal cycle of $CO_2$ concentration due to vegetation activities was clear, the assimilation window was ± 1 h and DA cycling was conducted at 6-h intervals. In previous studies that optimized $CO_2$ concentrations and carbon fluxes by assimilating $CO_2$ concentration observations, the cycling interval was 6 h or 24 h (Kang et al., 2011, 2012; Zhang et al., 2021; Liu et al., 2019; Liu et al., 2022)."

[Figure]

Figure_rev1_2. Time series of ensemble standard deviation of (a) 10 m U [m s$^{-1}$], (b) 10 m V [m s$^{-1}$], and (c) 2 m temperature [K], averaged over the four DA experiments.

*6. [line 354 (minor comment)] The authors provide many values with the unit $10^4$ ppm$^2$, to be interpreted as the sum of squared errors in all grid cells. A (very) brief guidance on how to interpret these numbers could possibly help the reader in section 3.3.1 and in Table 5.*

**Authors' response:** As the reviewer indicated, $e_a$, $e_b$, and nonlinear error forecast reduction (NER) in Section 3.3.1 were calculated as the sum of squared errors in all grid points. Previously, the NER unit was stated as $10^5$ J kg$^{-1}$ in Kim and Kim (2024) which investigated the effect of observations on meteorological forecast errors in the Arctic.

Table_rev1_2 shows the reference $e_{ref}$ value which is calculated assuming uniform forecast errors of "X" ppm occur across the entire grid. For instance, if there is an error of 1 ppm for all grids, the $e_{ref}$ would be 12.42 x $10^4$ $ppm^2$; for 5 ppm error, the $e_{ref}$ would be 310.38 x $10^4$ $ppm^2$. These reference values are useful for interpreting the magnitudes of $e_a$, $e_b$, and NER in Table 5.

As the reviewer suggested, we have added the following text under Table 5 in the revised manuscript: "Note: For reference, if a uniform forecast error of 1 ppm is assumed for all grid points, then the corresponding forecast error value is 12.42 x $10^4$ $ppm^2$."

Table_rev1_2. Reference forecast error ($e_{ref}$) values corresponding to assumed uniform forecast errors of "X" ppm across all grid points.

| Assumed uniform forecast error (ppm) | Reference forecast error ($e_{ref}$) |
|---|---|
| 1 ppm | 12.42 x $10^4$ $ppm^2$ |
| 2 ppm | 49.66 x $10^4$ $ppm^2$ |
| 3 ppm | 111.74 x $10^4$ $ppm^2$ |
| 4 ppm | 198.64 x $10^4$ $ppm^2$ |
| 5 ppm | 310.38 x $10^4$ $ppm^2$ |

*7. [lines 354 and 361 (minor comment)] The authors provide many numbers and even lists of numbers in the text. I do not want to criticize what is a matter of style, but instead of reading lists of 8 numbers as in lines 354 and 361, I personally prefer the structure of a table. The paper contains many tables already and some of the numbers provided in the text could be included in the existing tables, e.g., by including averages in Table 5.*

**Authors' response:** Following the reviewer's suggestion, we have moved the average values that were listed in the original manuscript into Table 5 and deleted the corresponding sentences. In addition to Table 5, we also added the average values for each experiment to Tables 4 and 6. The deleted parts are as follows.

Line 395-397 in the revised manuscript: "In all experiments and forecast times, $e_a$ was smaller than $e_b$, which implied that the forecast error of the $CO_2$ concentration was reduced with DA of surface $CO_2$ observations (Figs. 7a and b).  The forecast error was reduced by an average of 17.0% by assimilating the surface $CO_2$ observations."

Line 401-402 in the revised manuscript: "Assimilating evenly distributed surface $CO_2$ observations reduced forecast errors in EXP1 and EXP4.  The forecast errors with and without DA in all experiments increased as the forecast time increased from 6 to 24 h (Figs. 7a and b)."

*8. [lines 424 to 484 (minor comment)] Section 3.3.3 mainly discusses three separate aspects: The correlation of EFSO with self-sensitivity and $CO_2$ concentration variability; the impact of vegetation types around the observation sites; and the fraction of beneficial observations. For the reader it might be easier to follow if section 3.3.3 is split into two or even three parts (e.g. start a new section 3.3.4 on line 468 for the fraction of beneficial observations).*

**Authors' response:** Following the reviewer's suggestion, Section 3.3.3 has been split into two parts: Section 3.3.3. Ensemble forecast sensitivity to observation and Section 3.3.4. Fraction of beneficial observations.

*9. [lines 468 to 484 (minor comment)] This paragraph could be written more concisely. For example, the following sentences seem redundant: "The average fractions of beneficial observations ... were 68.9% and 66.3% ... On average, more than half of the observations contributed to reducing the forecast errors. ... Therefore, on average, more than half of the assimilated surface $CO_2$ observations in the four experiments contributed to the reduction of forecast errors."*

*In general, the presentation of the results is clear and understandable. But as the example shows, it could be more concise in a few paragraphs.*

**Authors' response:** Following the reviewer's suggestion, we have revised the paragraph to remove redundant sentences. The deleted parts are denoted.

Line 531-532 in the revised manuscript: "The average fractions of beneficial observations for the four experiments were 68.9% and 66.3% for the 6 h and 12h forecast, respectively. "

**Technical corrections:**

*1. [line 104] Eq. (4) I do not understand how Eq. (4) leads to $A^T$ in Eq. (3) (instead of just A).*

**Authors' response:** The ensemble adjustment Kalman filter (EAKF) equations in Section 2.2 are referred from Anderson (2001), which first introduced the EAKF method. The

EAKF in Data Assimilation Research Testbed (DART) system used in this study was developed based on the formulations in Anderson (2001, 2003).

Anderson (2001) addressed the limitations of the traditional Ensemble Kalman Filter (EnKF), particularly the spurious sampling noise introduced by perturbing observations. To overcome this, he proposed the Ensemble Adjustment Kalman Filter (EAKF), a novel approach that adjusts the ensemble deterministically. In the EAKF, ensemble perturbations are first projected onto the observation space, rescaled according to the Kalman update, and then mapped back to the model state space. This approach eliminates the need for stochastic perturbations of observations, reduces sampling noise, and provides more accurate analysis estimates, especially for small ensemble sizes.

Anderson (2001) used $\mathbf{A^T}$, while Anderson (2003) used $\mathbf{A}$ in Eq. (3), which implies that the choice between $\mathbf{A}$ and $\mathbf{A^T}$ is not essential. In practice, Eq. (3) is implemented in a scalar formulation, as shown in Anderson (2003). Many EAKF studies have used $\mathbf{A}$ and $\mathbf{A^T}$ interchangeably, and as noted above, they are equivalent in practical coding.

We have revised the text as follows. The revised parts are underlined.

Line 142-144 in the revised manuscript: "$\mathbf{A}$ in Eq. (3) satisfies the relationship given in Eq. (4). Using Eq. (4), $\mathbf{P}^a$ is adjusted to match the theoretical analysis error covariance in Eq. (2). More detailed formulations of the EAKF can be found in Anderson (2001)."

*2. [Fig. 1] The figure is very illustrative, but on the horizontal axis, the label "t h" seems odd to me since t is usually considered a value including a unit.*

**Authors' response:** Following the reviewer's suggestion, we have revised the Fig. 1 accordingly.

*3. [line 284] I was confused by the word "multiple", which is meant here in the sense of "too many".*

**Authors' response:** Following the reviewer's suggestion, we have revised the text as follows. The revised parts are underlined.

Line 321-322 in the revised manuscript: "When an observation error is not considered, a large number of observations can be counted in the first and last ranks."

*4. [line 381] "forecast reduction" should probably be "forecast error reduction".*

**Authors' response:** We have revised the text. The revised parts are underlined.

Line 418-420 in the revised manuscript: "The impact of the surface $CO_2$ observations added in EXP3 was large in the analysis (Fig. 5c), whereas it was relatively small in forecast error reduction."

5. ***[line 481]*** *Sites "A1, A4" should probably be "V1, V4".*

**Authors' response:** We have corrected the typo. The revised parts are underlined.

Line 538-540 in the revised manuscript: "More than half of the observations at Sites V1, V4, and R14 did not contribute to reducing forecast errors for the 12-h forecast, whereas the average EFSO impact of Sites V1, V4, and R14 negatively contributed to reducing the forecast error."

**References**

Anderson, J. L.: An ensemble adjustment Kalman filter for data assimilation, Mon. Weather Rev., 129(12), 2884-2903, doi: 10.1175/1520-0493(2001)129<2884:AEAKFF>2.0.CO;2, 2001.

Anderson, J. L.: A local least squares framework for ensemble filtering, Mon. Weather Rev., 131(4), 634-642, doi:10.1175/1520-0493(2003)131<0634:ALLSFF>2.0.CO;2, 2003.

Hilton, T. W., Davis, K. J., Keller, K., and Urban, N. M.: Improving North American terrestrial $CO_2$ flux diagnosis using spatial structure in land surface model residuals, Biogeosciences, 10(7), 4607-4625, doi:10.5194/bg-10-4607-2013, 2013.

Kim, D.-H., and Kim, H. M.: Adjoint-based observation impact on meteorological forecast errors in the Arctic, Q. J. R. Meteorol. Soc., 150, 5403-5421, doi: 10.1002/qj.4876, 2024.

Kim, H. M., and Kim, D.-H.: Effect of boundary conditions on adjoint-based forecast sensitivity observation impact in a regional model, J. Atmos. Ocean. Technol., 38, 1233-1247, doi:10.1175/JTECH-D-20-0040.1, 2021.

Li, X., Hu, X.-M., Cai, C., Jia, Q., Zhang, Y., Liu, J., Xue, M., Xu, J., Wen, R., and Crowell, S. M. R.: Terrestrial $CO_2$ fluxes, concentrations, sources and budget in Northeast China: Observational and modeling studies, J. Geophys. Res.-Atmos., 125, e2019JD031686, doi:10.1029/2019JD031686, 2020.

Seo, M.-G., Kim, H. M., and Kim, D.-H.: Effect of atmospheric conditions and VPRM parameters on high-resolution regional $CO_2$ simulations over East Asia, Theor. Appl. Climatol., 155, 859-877, doi:10.1007/s00704-023-04663-2, 2024.

Seo, M.-G., and Kim, H. M.: Evaluation of high-resolution regional $CO_2$ data assimilation-forecast system in East Asia using observing system simulation experiment and effect of observation network on simulated $CO_2$ concentrations, Q. J. R. Meteorol. Soc., e4987, doi:10.1002/qj.4987, 2025.

Yang, E.-G., Kim, H. M., Kim, J., and Kay, J. K.: Effect of observation network design on meteorological forecasts of Asian dust events, Mon. Weather Rev., 142, 4679-4695, doi:10.1175/MWR-D-14-00080.1, 2014.

**Egusphere-2025-2367 (Editor: Christoph Gerbig)**

**Response to Reviewer 2**

The authors thank the reviewer 2 for a thoughtful review of the manuscript. Considering the reviewer's points, we have made the necessary changes. The responses for the reviewer's specific comments are as follows.

**Comments:**

*1. The manuscript "Ensemble-based observation impact of surface $CO_2$ concentration observations on analysis and forecast of atmospheric $CO_2$ concentrations over East Asia" of Min-Gyung Seo and Hyun Mee Kim analyze the impact of $CO_2$ observations in OSSEs for the regional analysis (via data assimilation) and forecast of atmospheric $CO_2$ concentrations in East Asia.*

*The introduction does not really state a clear objective for such an analysis of the observation impact for existing and potential $CO_2$ observation sites (in particular, something is missing in lines 73-74) and lines 211-214 narrow down the potential scope of the study. However, in principle, it should support the extension of surface networks "to better analyze and forecast atmospheric $CO_2$ concentrations in East Asia" (l74).*

> **Authors' response:** The objective of this study was to investigate the impacts of surface $CO_2$ concentration observations on analysis after data assimilation (DA) and on $CO_2$ concentration forecasts to improve the analysis and forecast of atmospheric $CO_2$ concentrations in East Asia, as mentioned in lines 73-74 in the original manuscript.
>
> For this purpose, the observation impact was evaluated in pseudo observations from existing surface $CO_2$ observation sites under an observation system simulation experiment (OSSE). Because the number of existing surface $CO_2$ observation sites is limited in East Asia, we additionally evaluated the potential benefits of adding and redistributing surface $CO_2$ observation sites. These evaluations help to clarify the characteristics of observation impacts and provides useful insights for anticipating the impact of assimilating aircraft and satellite observations in future studies, as well as for designing improved observation networks to better analyze and forecast atmospheric $CO_2$ concentrations in East Asia.
>
> We agree that the objective and possible extensions of the study were not clearly stated in the introduction. To address this, the text in lines 211-214 in the original manuscript was combined with the objective in the introduction, and revised accordingly. Following the reviewer's suggestion, we have revised the manuscript to present the objective and its potential contributions more clearly. We have revised the text in Section 1 (line 73-74 in

the original manuscript) and Section 2.4.2 (line 211-214 in the original manuscript) as follows. The revised parts are underlined.

Line 79-89 in the revised manuscript: "In this study, the impacts of surface $CO_2$ concentration observations on analysis after DA and on $CO_2$ concentration forecasts were investigated, under various observation networks including existing observation network, to enhance the analysis and forecast of atmospheric $CO_2$ concentrations in East Asia. Using an observation system simulation experiment (OSSE), the impacts of surface $CO_2$ concentration observations were evaluated for pseudo observations from existing surface $CO_2$ observation sites. Compared to North America and Europe, the number of surface $CO_2$ observations in East Asia is insufficient, and in situ observations are even fewer. Investigation of the observation impact using the existing in situ observations has limitations in fully understanding the characteristics of the observation impact. Therefore, the potential benefits of adding and redistributing surface $CO_2$ observation sites were also examined. These evaluations help to understand the characteristics of observation impacts and provide insights for anticipating the effects of assimilating aircraft and satellite observations, as well as for designing future observation network to better analyze and forecast atmospheric $CO_2$ concentrations in East Asia."

Line 261-263 in the revised manuscript: " To examine the impact of each observation used for DA on the simulated $CO_2$ concentrations under the framework of existing in situ surface observation sites and changed observation sites, four experiments with different observation networks were conducted in this study."

*2. The authors deploy complex experimental, data assimilation and analysis theoretical and practical frameworks to conduct this study. However, as explained below, many major aspects of the study and of the manuscript raise concerns, from the rationale and objective of the study to the implementation of the experiments and interpretation of the results, including the quality of the writing. These concerns are such that they can hardly be addressed if limiting the request of the journal to major revisions, which is why I suggest to reject this manuscript and to encourage the authors to reconsider and rebuild their study before resubmitting a new manuscript based on their valuable tools.*

*From my point of view, the regional analysis and forecast of the atmospheric $CO_2$ concentrations is not a sensible target for the deployment of surface $CO_2$ networks. There is no major scientific or societal need for accurate forecasts of the $CO_2$ concentrations over short timescales. Global $CO_2$ forecasts at relatively high spatial resolution are often used to*

*constrain the boundary conditions of regional and local $CO_2$ atmospheric inversion systems (solving for the surface $CO_2$ fluxes), but in many cases, such regional and local inversion systems are coupled to global $CO_2$ inversion systems (solving for the surface fluxes). One could still argue that regional forecasts could also be used to constrain the boundary conditions of local atmospheric inversion systems. However, given the current lack of $CO_2$ surface stations, it would not make sense to optimize the design of relatively dense continental monitoring networks for such an objective. In any case, the analysis and discussions in this study do not provide indications regarding the potential for such a coupling.*

*The introduction is quite revealing regarding this concern: the part dedicated to the rationale of the study up to line 36 mainly discusses the need to estimate the $CO_2$ fluxes. However, line 37 jumps into the analysis and forecast of $CO_2$ concentrations without explanations, starting with a "Therefore", which artificially connects the two parts.*

**Authors' response:** Atmospheric $CO_2$ concentration is a fundamental variable measured by various observation platforms, including surface stations, aircraft, and satellites. The model's ability to simulate atmospheric $CO_2$ concentrations reflects how well the model represents reality. In addition to the use of $CO_2$ concentration forecasts for boundary conditions of the regional model, mentioned by the reviewer, accurately simulating $CO_2$ concentrations in the model is important from the following two perspectives.

First, the accuracy of optimized $CO_2$ fluxes in inverse modeling is validated not by comparing optimized $CO_2$ fluxes directly with observed $CO_2$ fluxes, but by comparing simulated $CO_2$ concentrations (using optimized fluxes) against observed $CO_2$ concentrations. For instance, Kim et al. (2017) and Cho and Kim (2022), which optimized $CO_2$ fluxes over East Asia using inverse modeling with the CarbonTracker, evaluated the accuracy of the optimized $CO_2$ fluxes by comparing the simulated $CO_2$ concentrations against observed $CO_2$ concentrations. Thus, high-resolution $CO_2$ concentrations simulated in the model can be used to validate the $CO_2$ fluxes from the inverse modeling.

Second, accurate analyses and forecasts of atmospheric $CO_2$ concentrations are necessary to produce accurate optimized $CO_2$ fluxes in the inverse modeling. Inaccurate $CO_2$ concentration forecasts lead to large discrepancies with observed $CO_2$ concentrations. This makes the flux optimization in inverse modeling difficult because the differences between them (i.e., $CO_2$ concentration observations and forecasts) are used to update the surface $CO_2$ fluxes by the data assimilation in the inverse modeling. Thus, $CO_2$ concentrations and surface $CO_2$ fluxes are augmented in the data assimilation, forward modeling, and inverse modeling system, which cannot be treated separately.

Accurate $CO_2$ concentration analysis and forecasts are therefore essential for accurate $CO_2$ flux estimates. For this reason, numerous previous studies have investigated the accuracy of $CO_2$ concentration simulations using regional atmospheric chemistry models (Chen et al., 2019; Díaz-Isaac et al., 2018; Dong et al., 2021; Gerken et al., 2021; Seo et al., 2024b).

The objective of this study was to investigate the impacts of surface $CO_2$ concentration observations on analysis after data assimilation and on $CO_2$ concentration forecasts, under various observation networks, to improve the analysis and forecast of atmospheric $CO_2$ concentrations in East Asia. By quantitatively evaluating the effect of observed $CO_2$ concentrations on the analysis and forecast of $CO_2$ concentrations, one can indirectly infer how the observed $CO_2$ concentrations affect the optimization of surface $CO_2$ fluxes.

For future studies, we plan to expand our system to inverse modeling system for optimizing surface $CO_2$ fluxes. To develop the inverse modeling system, the analysis and forecast of $CO_2$ concentrations should be validated and the effects of assimilated observations on the analysis and forecast of $CO_2$ concentrations need to be fully understand. In addition, although the current surface $CO_2$ observation network in East Asia is sparse, advances in low-cost sensors and portable, high-precision instruments (Martin et al., 2017; Shusterman et al., 2018) make the construction of a denser network feasible in the future. By assessing the potential benefits of additional $CO_2$ observation sites, this study provides scientific evidence to support investments in expanding $CO_2$ observation infrastructure.

Following the reviewer's suggestion, the manuscript has been revised to clarify the logical connection between the $CO_2$ concentration simulation and estimation of surface $CO_2$ fluxes. The revised parts are underlined.

Line 24-44 in the revised manuscript: "Atmospheric $CO_2$ concentrations are influenced by various factors, including $CO_2$ emissions from several sources, respiration and photosynthesis of vegetation, and interactions between the atmosphere and the oceans (Ussiri and Lal, 2017). Efforts have been made to reduce the recently elevated atmospheric $CO_2$ concentrations (Le Quéré et al., 2019; Zhang et al., 2020). To verify whether efforts to reduce $CO_2$ emissions have been implemented, the distribution of the surface $CO_2$ flux has to be accurately estimated. Many studies have attempted to optimize surface $CO_2$ fluxes using inverse modeling by combining atmospheric chemistry models with data assimilation (DA) (Kim et al., 2014a, 2014b; Kim et al., 2017a; Kim et al., 2018; Monteil et al., 2020; Park and Kim, 2020; Wang et al., 2020; Maksyutov et al., 2021; Zhang et al., 2021; Cho and Kim, 2022). Accurate simulations of atmospheric $CO_2$ concentrations are essential for estimating optimized surface $CO_2$ fluxes in inverse modeling, since discrepancies between simulated and observed $CO_2$ concentrations directly constrain surface $CO_2$ flux estimation through DA. Thus, atmospheric $CO_2$ concentrations and surface $CO_2$ fluxes are augmented in the DA, forward modeling, and inverse modeling frameworks, and cannot be treated independently. For this reason, various studies have investigated the accuracy of $CO_2$ concentration simulations using regional atmospheric chemistry models (Chen et al., 2019; Díaz-Isaac et al., 2018; Dong et al., 2021; Gerken et al., 2021; Seo et al., 2024b). In previous studies that optimized the surface $CO_2$ flux and atmospheric $CO_2$ concentration using modeling, surface $CO_2$ flux observations were not used, but atmospheric $CO_2$ concentration observations were used in the DA. Atmospheric $CO_2$ concentration observations include surface in situ and flask observations, aircraft

observations, and satellite observations. The characteristics of the $CO_2$ concentration observations from various sources differ (Byrne et al., 2017). In particular, the number of $CO_2$ concentration observations in East Asia, which is important for understanding the carbon cycle, is smaller than those in North America and Europe (Byrne et al., 2017; Seo and Kim, 2023; Seo et al., 2024a, b). Therefore, understanding how each observation reduces the analysis and forecast errors of $CO_2$ concentration estimation is crucial for optimizing atmospheric $CO_2$ concentrations using DA with limited atmospheric $CO_2$ concentration observations."

*3. Even if assuming that optimizing the design of continental networks as a function of their skill for supporting the analysis and forecast of $CO_2$ concentrations could make sense, the study keeps on raising concerns.*

*The analysis for each assimilation window focuses on the correction of the $CO_2$ initial conditions, without correcting the surface fluxes. However, even over few hours, $CO_2$ concentrations are highly impacted by such surface fluxes, and in particular (when considering networks such as those tested here) by the surface land ecosystem fluxes. In the real world, the system that is tested here would thus project large biases in the analysis of the $CO_2$ atmospheric fields to compensate for the large uncertainties arising from the surface fluxes during the assimilation window, and the forecasting skill of such a system would be strongly limited by ignoring these fluxes in the analysis. This point is missed by the experiments here, because the authors implicitly assume that the biogenic surface fluxes are perfectly known (the true and "perturbed" estimate of these fluxes are identical = the outputs from VPRM). This issue may not be so significant if there had not been a large number of regional scale atmospheric inversion systems and studies in the past decades.*

**Authors' response:** As mentioned in the authors' response for comment 1 above, the objective of this study was to investigate the impacts of surface $CO_2$ concentration observations on analysis after data assimilation (DA) and on $CO_2$ concentration forecasts to improve the analysis and forecast of atmospheric $CO_2$ concentrations in East Asia. For this purpose, the observation impact was evaluated for existing real network and additional networks for possible extension. The optimization of observing networks is not the main purpose of this study.

It seems that mentioning simply "VPRM" for biogenic emission in the original manuscript may have caused some misunderstanding. In this study, to reflect the uncertainty of each emission and to avoid the identical twin problem in OSSE, different emission data were used for true state and DA experiments (Table_rev2_1). In VPRM simulations of biogenic $CO_2$ in WRF-Chem, empirical parameters $\alpha$, $\beta$, $\gamma$, and $PAR_0$ should be optimized for each land use type in the experimental region (Hilton et al. 2013). Although VPRM module was used to simulate biogenic $CO_2$ fluxes in true state and DA experiments, four

parameters ($\alpha$, $\beta$, $\gamma$, and $PAR_0$) in VPRM were different in true state and DA experiments, to provide sufficient differences in the biogenic $CO_2$ concentrations in true state and DA experiments. In this study, based on Seo et al. (2024a), which analyzed the effect of VPRM parameters on $CO_2$ simulations over East Asia, the US parameter table and Li parameter table (Li et al. 2020) were used for true state and DA experiments, respectively.

Figure_rev2_1 shows the distributions of anthropogenic and biogenic $CO_2$ concentrations simulated in true state and DA experiments, along with the spatial distribution of absolute relative differences of DA experiments from the true state. The distribution of DA experiments is the average distribution of four DA experiments. The average difference in anthropogenic $CO_2$ concentrations between true state and DA experiments is 2.7%, and that of biogenic $CO_2$ concentrations is 2.2%, indicating that the biogenic $CO_2$ concentrations in true state and DA experiments are not identical and differ similar magnitudes as the anthropogenic $CO_2$ concentrations.

Table_rev2_1. Description of emission data used in each experiment.

| Experiments | Emission input | | | DA |
| --- | --- | --- | --- | --- |
| | Anthropogenic | Biogenic | Oceanic | |
| True state | Average of CT2022 and ODIAC | VPRM * US table | CT2022 | X |
| DA experiments (EXP1, EXP2, EXP3, and EXP4) | ODIAC | VPRM * Li table | JMA ocean map | O |

[Figure]

Figure_rev2_1. Distribution of average anthropogenic $CO_2$ concentrations (ppm) in (a) true state and (b) four DA experiments, and average biogenic $CO_2$ concentrations (ppm) in (d) true state and (e) four DA experiments. Absolute relative difference (%) distribution between DA experiments and true state based on true state for (c) anthropogenic and (f) biogenic $CO_2$ concentrations.

To clarify the differences in the VPRM parameter tables used in each experiment, we have revised the text as follows. The revised parts are underlined.

Line 286-294 in the revised manuscript: "The emission data used in the four experiments differed from those used to simulate the true state. This is to avoid the identical twin problem that can occur in the OSSE by setting the experimental design of the true state and four experiments sufficiently different (Masutani et al., 2010; Shu et al., 2023; Kim et al., 2022). In the four experiments, ODIAC v2020b (Oda and Maksyutov, 2015) was used for anthropogenic emission, the Japan Meteorological Agency (JMA) ocean map (Iida et al., 2021; Takatani et al., 2014) was used for oceanic emission, and VPRM was used for biogenic emission. To ensure sufficient differences between the biogenic $CO_2$ in the true state and in the four experiments, different VPRM parameter tables were used: the US table for the true state and the Li table (Li et al., 2020) for the four experiments. Seo et al. (2024a) showed that biogenic $CO_2$ fluxes in East Asia vary considerably depending on the VPRM

parameter tables used, and the parameter values for the US and Li tables are described in detail in Seo et al. (2024a)."

*4. Furthermore, even though they use complex diagnostics to read the results from their experiments, the authors face difficulties to interpret them:*

*- a challenge associated to such a study is the mix between the observation impact of a given station within a given network, which can be very different within another network, and the impact of using different networks. The experimental framework and the analysis here do not fully ensure the distinction between these two impacts which limits the ability to draw robust conclusions regarding specific types of stations or of networks.*

**Authors' response:** Following the reviewer's suggestion, we examined whether the effects of observations of a given station within a given network and that within different networks are mixed. We have compared the self-sensitivity and EFSO impact for the overlapping observation sites in multiple experiments.

1) Self-sensitivity

Self-sensitivity is a measure of how much observation information is reflected in generating the analysis in DA. Specifically, self-sensitivity quantifies how much the analysis at a given observation site changes due to the assimilated observation, considering only the diagonal component of the influence matrix (Eq. 7 in the manuscript). Table_rev2_2 presents the self-sensitivity for sites that overlapped in two or more experiments. As shown in Table_rev2_2, the self-sensitivity for a given observation site was very similar across different experiments, indicating that the influence of each observation on the analysis is not significantly affected by the surrounding observation network.

According to the self-sensitivity equation (Eq. 7), self-sensitivity is fundamentally proportional to the ratio of the analysis error to the observation error at a given site. In our OSSE framework, the observation error was set to be identical for all sites, making the magnitude of the analysis error the primary factor determining self-sensitivity. As analyzed in Figure 6 of the original manuscript, self-sensitivity was strongly positively correlated with the analysis RMSE at each observation site. Therefore, self-sensitivity reflects the unique characteristics of the observation location, while the influence of the observation networks is relatively small and limited.

Table_rev2_2. Average self-sensitivity (%) at overlapped observation sites in EXP2, EXP3, and EXP4.

| Site ID | EXP2 | EXP3 | EXP4 |
|---------|------|------|------|
| YON | 5.1 | 5.0 | 5.0 |
| AMY | 27.0 | 26.8 | 26.8 |
| DDR | 19.1 | 19.0 | 19.0 |
| KIS | 18.1 | 17.9 | 18.0 |
| RYO | 9.6 | 9.7 | 9.6 |
| V1 | - | 52.2 | 51.1 |
| V2 | - | 25.9 | 26.8 |
| V3 | - | 36.7 | 38.4 |

2) EFSO impact

The EFSO impact quantifies how much an observation reduces the forecast error across the entire model domain, and thus its characteristics differ from those of self-sensitivity. The analysis and background used in the EFSO calculation result from integrating information from all assimilated observations, rather than from a single observation alone. As a result, the observation impact on forecast error reduction from each observation site can vary depending on its interaction with other observations in the network.

For example, when observations are dense in a specific region, some of the information may be redundant, leading to a reduced EFSO for an individual site (Casaretto et al., 2023). Table_rev2_3 shows the EFSO impact for the 6-h and 12-h forecasts at the overlapping observation sites. Unlike self-sensitivity, the EFSO impact for the same observation sites varied considerably depending on the observation network.

Table_rev2_3. Average EFSO impact at overlapping observation sites in EXP2, EXP3, and EXP4. The unit is $\times 10^4$ ppm$^2$.

| Site ID | 6 h forecast | | | 12 h forecast | | |
|---------|------|------|------|------|------|------|
| | EXP2 | EXP3 | EXP4 | EXP2 | EXP3 | EXP4 |
| YON | 0.00 | -0.03 | 0.00 | 0.00 | -0.01 | 0.00 |
| AMY | -0.64 | -1.82 | -1.18 | -1.94 | -3.96 | -1.12 |
| DDR | -0.40 | -0.28 | -0.52 | -0.42 | -0.29 | -0.28 |

| | | | | | | |
|---|---|---|---|---|---|---|
| KIS | -0.47 | -0.05 | 0.14 | -0.71 | -0.49 | -0.12 |
| RYO | -0.48 | -0.47 | -0.43 | -0.22 | -0.29 | -0.22 |
| V1 | - | -4.73 | -7.62 | - | -3.40 | -4.66 |
| V2 | - | -3.91 | -2.44 | - | -4.44 | -8.63 |
| V3 | - | -3.70 | -7.34 | - | -5.69 | -6.65 |

To analyze the impact of a single observation on forecast error reduction without the influence of nearby observations, one could ideally conduct single observation experiments or increase the distance between observation sites greatly. However, because single observation experiments are computationally expensive and unrealistic, many studies employ the EFSO method to efficiently assess observation impacts given network environment (Chang et al., 2023; Gasperoni et al., 2024). Thus, the mixed impact of EFSO is an inherent characteristic of the EFSO method rather than a limitation.

To clarify the different characteristics of self-sensitivity and EFSO impact, we have revised the text as follows. The revised parts are underlined.

Line 352-358 in the revised manuscript: "In each assimilation cycle, more than half of the observational information from V1 site was reflected in the analysis. For the existing observation sites in EXP2, AMY, XL, and HF showed larger self-sensitivity. These sites coincided with regions of large hourly $CO_2$ variability identified by the Variability strategy (Fig. 1b). For the overlapping observation sites (e.g., YON, AMY, DDR, KIS, RYO in EXP2, EXP3, and EXP4, and V1, V2, and V3 in EXP3 and EXP4), the self-sensitivity was similar regardless of the network configuration (Fig. 5 and Table 4). This indicates that self-sensitivity is more influenced by the intrinsic characteristics of each observation site rather than by the surrounding observation network design, which can be inferred from Eq. (7)."

Line 483-489 in the revised manuscript: "This implies that the impact on reducing forecast errors was greater when the observation sites were located evenly over the entire domain considering the variability of $CO_2$ concentrations, rather than simply randomly locating observation sites throughout the entire domain.

The EFSO impact exhibited different characteristics from self-sensitivity. While self-sensitivity for a specific site remained consistent regardless of the surrounding observation network (Table 4), the EFSO impact varied depending on the observation network (Table 6). This indicates that the EFSO impact does not measure the inherent effect of an individual site but quantifies its contribution to forecast error reduction within a specific network, influenced by other observation sites."

*- the lack of account for the atmospheric transport conditions over East Asia in July 2019 when evaluating the impact of the different stations or networks is an issue, e.g., since the positioning of the stations with respect to the study domain, to each other or to the domain boundaries when following the wind probably plays an important role in the forecasting skills, and since the transport conditions in July 2019 could be specific.*

**Authors' response:** The objective of this study is to quantitatively evaluate the effect of individual $CO_2$ observations on the analysis and forecasts of $CO_2$ concentrations within an OSSE framework. This objective differs from trying to perfectly reproduce actual $CO_2$ concentrations by assimilating real $CO_2$ concentration observations.

July was chosen as the experimental period because it represents a time when uncertainties from both anthropogenic and biogenic emission sources, which affect the uncertainty of $CO_2$ concentrations, are appropriately reflected. In contrast, experiments for winter period would have been dominated by uncertainties from anthropogenic emissions alone. Therefore, the experiment was conducted for the one-month period of July, to reflect comparable contributions from both anthropogenic and biogenic emission sources and to comprehensively evaluate the effects of surface $CO_2$ observations on the reduction of analysis and forecast errors of $CO_2$ concentrations.

We examined whether the meteorological conditions during the experimental period, July 2019, were anomalous compared to the 30-year (i.e., 1991-2020) climatological mean from the European Centre for Medium-Range Weather Forecast (ECMWF) Reanalysis v5 (ERA5) reanalysis. The meteorological conditions for July 2019 simulated in WRF-Chem were compared with the climatological mean at the 500 hPa and surface level. Specifically, the geopotential height and horizontal wind were compared at 500 hPa, while mean sea level pressure (MSLP) and horizontal wind were compared at the surface level (Figure_rev2_2). At 500 hPa, the geopotential height pattern in July 2019 closely resembled the climatological mean, and the position of the edge of the North Pacific High was very similar in both July 2019 and climatological mean (Figure_rev2_2a, b). At the surface, the location of the North Pacific High and mean horizontal wind in July 2019 were also nearly identical to the climatological mean, showing a southerly flow (Figure_rev2_2c, d). These results indicate that the meteorological conditions in July 2019 were not significantly different from the climatological mean, confirming that the experiment was conducted under representative conditions of July.

In addition, to consider the effect of atmospheric transport on the calculation of observation impacts, a moving localization method was applied in the EFSO calculation. This method shifts the localization center based on the horizontal wind forecast at each time step, thereby reflecting the influence of the simulated winds on the calculation of observation impacts.

To clarify the reason for selecting July as the experimental period, we have revised the text as follows. The revised parts are underlined.

Line 116-120 in the revised manuscript: "The experimental period was from June 22 to July 31, 2019, and the spin up period for model stabilization was from June 22 to June 30, 2019. This period was selected to reflect uncertainties from both anthropogenic emissions and biogenic fluxes. A comparison with the 30-year (1991-2020) climatological mean from the ERA5 reanalysis confirmed that the large-scale meteorological conditions in July 2019 were representative of a typical summer in East Asia (not shown). The experimental domain was East Asia with a horizontal resolution of 9 km and 51 vertical layers (Fig. 1)."

[Figure]

Figure_rev2_2. The average 500 hPa geopotential height [m] (black line) and 500 hPa wind [m s$^{-1}$] (blue vector) in (a) July 2019 from WRF-Chem and (b) climatological mean from 30-year (i.e., 1991-2020) of ERA5 reanalysis. The average mean sea level pressure [hPa] (contour) and surface wind [m s$^{-1}$] (blue vector) in (c) July 2019 from WRF-Chem and (d) climatological mean from 30-year of ERA5 reanalysis.

*- the lack of clear information on and characterization of the initial and/or sequential derivation of the uncertainties in the background state / initial conditions (the Pb matrix and the spread of the (xb)_i) of the assimilation windows (what are the spatial scales of the correlations associated to these uncertainties?) further limits the ability to analyze properly the observation impacts (page 4 is confusing; the discussions on min distances between stations = 600 km or 300 km on page 9 is highly questioning).*

**Authors' response:** We have addressed the reviewer's questions as follows.

1) Characteristics of uncertainties in the background state/initial conditions.

The $CO_2$ DA-forecast system used in this study was first developed and fully evaluated in Seo and Kim (2025) through various analyses. The configuration of model and data assimilation in this study is identical to that of Seo and Kim (2025). An analysis of the uncertainties in the background state/initial conditions was sufficiently discussed in Seo and Kim (2025). To avoid redundancy, only the rank histogram was analyzed in this study. However, following the reviewer's suggestion, we additionally evaluated the uncertainties of background state in the DA experiments of this study to clarify them further.

In an ensemble DA system, one way to diagnose the uncertainties in the background state is to compare the ensemble spread with root mean square error (RMSE) between the background state interpolated to the observation space and the observations. If the ensemble DA system operates stably throughout the experimental period, the RMSE and total spread are expected to have comparable magnitudes (Raeder et al., 2012; Jung et al., 2012).

Figure_rev2_3 shows the time series of RMSE and total spread for the four DA experiments (i.e., EXP1, EXP2, EXP3, and EXP4) conducted in this study. Although the RMSE was slightly larger in EXP3 and EXP4, which included observation sites with high variability, compared to EXP1 and EXP2, the trends and magnitudes of the RMSE and total spread were similar across all experiments. This indicates that the ensemble DA system used in this study adequately represents the uncertainties in the background state.

Furthermore, the background error covariance in the ensemble adjustment Kalman filter (EAKF) is flow-dependently determined by 20 ensembles. To reduce the spurious correlations that may result from this limited ensemble size, spatial localization with a radius of 1274.2 km was applied. Therefore, the effective spatial correlation scales in the DA are determined by physically meaningful correlations at distance shorter than the localization radius.

We have added the text in Section 2.1 as follows. The revised parts are underlined.

Line 107-109 in the revised manuscript: "The CO$_2$ DA-forecast system used in this study was first developed and comprehensively evaluated in Seo and Kim (2025). Detailed descriptions of the model and data assimilation configurations and extensive analyses of system stability are provided in Seo and Kim (2025)."

[Figure]

Figure_rev2_3. Time series of RMSE [ppm] (black solid line) and total spread [ppm] (gray dashed line) of surface CO$_2$ concentrations in (a) EXP1, (b) EXP2, (c) EXP3, and (d) EXP4 with respect to the pseudo CO$_2$ concentrations.

2) Discussion on the minimum distance between observation sites.

In an ensemble DA system, spatial localization is applied to reduce spurious correlations arising from the limited number of ensemble members. In this study, a localization radius of 1274.2 km was used for surface CO$_2$ observations. Theoretically, if the distance between observation sites is sufficiently greater than the localization radius, the impact of each observation on the analysis and forecast error reduction can be considered independent.

However, in the actual CO$_2$ observation network over East Asia, many CO$_2$ observation sites are located closer than the localization radius. Moreover, designing a realistic observation network requires considerations beyond statistical independence, including cost and monitoring of specific emission sources. Therefore, this study aimed to analyze observation impacts under realistically achievable networks with different densities and objectives, rather than assuming ideal statistical independence.

Yang et al. (2014), which conducted observation network experiments under an OSSE framework for Asian dust forecasts, showed that selecting random sites with greater distances between sites (up to 300 km) more effectively reduced forecast errors. In addition, adding observations in sensitive regions based on adjoint-sensitivity reduced forecast error more than locating them randomly. However, if observation sites were too dense within these sensitive regions, the performance in forecast error reduction could be reduced.

Based on Yang et al. (2014), minimum distances between observation sites were determined for two strategies (i.e., the Random and Variability strategies). For the Random strategy, a minimum distance of 600 km was used to distribute observation sites as evenly as possible over the land of the experimental domain, while matching the number of observation sites (i.e., 13) with the number of real observation sites in EXP2. For the Variability strategy, a minimum distance of 300 km was used to simulate a high-density observation network for targeted monitoring of regions with high surface $CO_2$ variability. Distances larger than 300 km would prevent placing observation sites in these high-variability regions. The minimum distance between observation sites of 300 km is also the distance that showed the most effective improvement in dust forecast performance in Yang et al. (2014).

Therefore, the minimum distance of 600 km for the Random strategy and 300 km for the Variability strategy are not arbitrary but represent observation networks that can be considered in reality.

We have added the reason for the selected minimum distances as follows. The revised parts are underlined.

Line 264-275 in the revised manuscript: "Two strategies (i.e., Random and Variability) were used to select surface $CO_2$ observation sites. The Random strategy randomly selects observation locations within the experimental domain, ensuring that the observation locations are at least 600 km from each other. This is because close observation locations are less effective for reducing the forecast error (Yang et al., 2014). A minimum distance of 600 km was used to distribute the 13 observation sites as evenly as possible over the land of the experimental domain, matching the number of real observation sites in EXP2."

The Variability strategy selects observation locations from regions with highly variable true $CO_2$ concentrations, which have high standard deviation of the true state at each grid point (Fig. 1b). In the Variability strategy, observation sites were selected starting from the grid with the greatest standard deviation, and the distances between the observation locations were at least 300 km from each other. Distances larger than 300 km would prevent placing observation sites in these high-variability regions. The 300 km distance for Variability strategy is the distance that showed the most effective improvement in dust forecast performance in Yang et al. (2014). For the Random and Variability strategies, the

selected observation sites were located on land and at least 10 grids away from each domain boundary."

*- The conclusion that stations located in areas with strong biogenic fluxes have a larger impact could be questioning since there is no perturbation of these fluxes in the experiments. The authors do not propose a mechanism to explain this. It could actually be linked to the transport, e.g. the PBL (a strong driver of the $CO_2$ diurnal variations that is ignored at lines 451-452), whose modeling scheme is perturbed in the OSSEs. In areas with high biogenic fluxes, such a transport uncertainty would propagate into higher $CO_2$ errors.*

**Authors' response:** As explained in our response to reviewer's comment 3, the biogenic $CO_2$ concentrations in the true state and DA experiments were simulated using different parameter tables in VPRM to account for uncertainties in biogenic fluxes. This resulted in a 2.2% difference in biogenic $CO_2$ concentrations, comparable to the 2.7% difference in anthropogenic $CO_2$ concentrations between the true state and the DA experiments.

Regions with active vegetation exhibit large diurnal variability in $CO_2$ concentrations, with large $CO_2$ uptake from photosynthesis during the daytime and large $CO_2$ release from respiration at nighttime. We consider uncertainties associated with large diurnal variability in $CO_2$ concentrations in active vegetation regions to be the primary source of uncertainty in this study. In addition, as the reviewer indicated, the errors related to transport and planetary boundary layer height (PBLH) could also influence uncertainties in $CO_2$ concentrations. In regions with large diurnal $CO_2$ variability, forecast errors may be amplified by uncertainties associated with transport and PBLH, under specific conditions.

Nevertheless, this study was conducted within an OSSE framework for a one-month period, in which the four DA experiments were simulated using the same transport model (i.e., WRF-Chem) with similar perturbations. As noted in Zheng et al. (2018), the impact of the systematic transport error is relatively small in this OSSE framework.

Therefore, the primary source of uncertainty in simulating $CO_2$ concentrations in this study was from the differences in emission inventories between the true state and the DA experiments. The effects of uncertainties in transport and PBLH simulations on $CO_2$ forecast errors are beyond the scope of this study and would be addressed in future work.

*5. The authors do not discuss any potential problem associated with the assimilation of $CO_2$ observations at night while most of the global to local inverse modelling systems keep on avoiding to assimilate nighttime $CO_2$ observations from plain or low altitude stations due to the large $CO_2$ transport modelling biases at night.*

**Authors' response:** In this study, surface pseudo $CO_2$ observations were assimilated at 6 h interval, including both daytime and nighttime observations. Previous studies have also

assimilated both daytime and nighttime observations: 1) Peng et al. (2023) developed a $CO_2$ inverse modeling system using The Community Multiscale Air Quality model (CMAQ) and ensemble Kalman smoother (EnKS), and assimilated real $CO_2$ observations from 14 surface sites in East Asia. $CO_2$ observations from all hours were used for DA except for 5 observation sites where only daytime observations were available. In regions where only daytime observations were assimilated, the optimized posterior carbon fluxes were underestimated because only observations from the daytime when photosynthesis is active were assimilated. 2) Ma et al. (2019), which simulated air pollutant using a WRF-Chem and data assimilation research testbed (DART) system, also assimilated all available observations from surface observation sites to analyze simulation performance for both day and night. As demonstrated in these previous studies, assimilating all available observations is effective to properly simulate the diurnal cycle, despite challenges in modeling the nocturnal boundary layer.

Furthermore, this study was conducted within an OSSE framework, in which the true state and the four DA experiments were simulated using the same transport model (i.e., WRF-Chem). As denoted in Zheng et al. (2018), the impact of the systematic transport error is negligible in this OSSE framework. Thus, the primary source of uncertainty in simulating $CO_2$ concentrations in this study was from the differences in emission inventories between the true state and the DA experiments.

To investigate the impact of assimilating nighttime observations, we categorized the daily cycle into two groups (i.e., daytime and nighttime). Based on the general PBL evolution (Figure 1.7 in Stull, 1988) in East Asia, we classified 00 UTC (growth phase of the convective mixed layer) and 06 UTC (peak of convective mixed layer) as daytime (convective regime), while 12 UTC and 18 UTC, when a stable layer forms due to surface cooling, were classified as nighttime (stable regime).

First, we examined the impact of excluding nighttime observations. If observations are assimilated continuously every 6 hour, the background for the 00 UTC cycle (the start of the daytime) is 6-hour forecast (Figure_rev2_4a). In contrast, when only daytime (00 and 06 UTC) observations are assimilated, the background for the 00 UTC cycle is 18-hour forecast (Figure_rev2_4b). Comparison of background RMSE at 00 UTC for these two cases shows a significant increase in error when nighttime observations are not assimilated (Figure_rev2_5). This increase was particularly pronounced in EXP3 and EXP4, which included observation sites with large diurnal $CO_2$ variability. Thus, omitting nighttime $CO_2$ observations in DA allows the uncertainties from the emission inventory accumulated and uncorrected during nighttime, which degrades the accuracy of the background forecasts for the following daytime.

**(a) DA using all observations**

[Figure]

**(b) DA using only daytime observations**

[Figure]

Figure_rev2_4. Schematic diagram for two DA cases: (a) a continuous DA with 6-h cycles using all observations, and (b) a hypothetical DA using only daytime observations. The yellow star represents the background at 00 UTC (i.e., the start of the daytime). The solid arrows represent the model forecast, and the dotted lines represent the analysis step including DA. Gray shaded boxes represent the nighttime (i.e., 12 and 18 UTC).

[Figure]

Figure_rev2_5. Background RMSE [ppm] at 00 UTC for each experiment in Figure_rev2_4 with respect to the pseudo $CO_2$ observations. All cycles DA (gray bar) represents the case with assimilation every 6 h, while the Daytime cycles DA (black bar) represents the hypothetical assimilation during only daytime (i.e., 00 and 06 UTC).

Next, we evaluated $CO_2$ simulation performance during daytime (i.e., 00 and 06 UTC) and nighttime (i.e., 12 and 18 UTC) (Figure_rev2_6a) for the experiments in this study. The background RMSE was larger during the nighttime in all experiments, reflecting greater uncertainties from emissions during nighttime. However, after assimilating surface $CO_2$ observations, the analysis RMSE during the nighttime was reduced considerably. Across assimilation cycles (Figure_rev2_6b), background RMSE was lowest at 06 UTC, but the impact of DA was significant at all hours. The RMSE reduction rate from background to analysis exceeded 30% for all experiments at all times (Table_rev2_4), indicating that the nighttime observations at 12 and 18 UTC, when uncertainty was greatest, also contributed substantially to improving $CO_2$ simulation performance. Therefore, assimilating nighttime observations is essential for maintaining overall accuracy and stability by most effectively improving model performance when its uncertainty is greatest.

[Figure]

Figure_rev2_6. (a) RMSE [ppm] of the background (Back) and analysis (Anal) for each experiment with respect to the pseudo $CO_2$ observations, averaged for daytime cycles (i.e., 00 and 06 UTC) and nighttime cycles (i.e., 12 and 18 UTC). (b) RMSE [ppm] of the background (Back) and analysis (Anal) for each experiment and for each assimilation cycle (00, 06, 12, and 18 UTC) with respect to the pseudo $CO_2$ observations.

Table_rev2_4. RMSE [ppm] of the background and analysis for each experiment and assimilation cycle, with respect to the assimilated pseudo $CO_2$ observations. RMSE reduction (%) of the analysis compared to the background.

| Exp. name | Time [UTC] | Background [ppm] | Analysis [ppm] | Reduction [%] |
|---|---|---|---|---|
| EXP1 | 00 | 4.95 | 2.93 | 40.7 |
|  | 06 | 3.44 | 1.90 | 44.9 |
|  | 12 | 4.54 | 2.68 | 40.9 |

| | 18 | 5.50 | 2.50 | 54.6 |
|---|---|---|---|---|
| EXP2 | 00 | 4.61 | 2.80 | 39.2 |
| | 06 | 3.84 | 2.56 | 33.2 |
| | 12 | 4.13 | 2.77 | 32.9 |
| | 18 | 4.54 | 2.41 | 46.9 |
| EXP3 | 00 | 10.59 | 7.11 | 32.8 |
| | 06 | 4.44 | 2.45 | 44.6 |
| | 12 | 9.71 | 4.43 | 54.4 |
| | 18 | 13.61 | 5.80 | 57.4 |
| EXP4 | 00 | 7.34 | 4.66 | 36.5 |
| | 06 | 4.46 | 2.56 | 42.6 |
| | 12 | 7.58 | 4.18 | 44.9 |
| | 18 | 8.92 | 4.47 | 49.8 |

*6. The writing of the manuscript is not satisfactory, many sections are unclear, and they do not introduce the main goals, concepts and ideas in a clean way, which does not encourage the reader to delve into the mathematical framework. As an example, the lines 105-117 which contain key information are confusing: here, the authors do not really try to describe or explain things rigorously but rather to list the values or options for their various input parameters. Part of the basic information on the study domain, period, modelling framework (content of the state vector x, spatial resolutions, duration of the assimilation windows and forecasts...) etc. is delivered too late, i.e., after considerations that should be driven by such information, or is simply skipped (see the discussion on Pb above). The abstract provides a first good illustration of this general problem. In the introduction and result sections, the authors often lose the reader with a high number of statements and statistics which do not systematically follow a logical flow or which do not seem to be the most relevant.*

**Authors' response:** Following the reviewer's opinion, we have revised Section 2 Methods. Specifically, the text in Section 2.5 was moved to Section 2.1, 2.3.1, and 2.3.2. In addition, as noted in the authors' responses above, Introduction, and Results sections were revised to improve the overall structure and logical flow of the manuscript.

The revised parts for Section 2 are as follows and underlined.

Line 105-122 in the revised manuscript: "In this study, DART was modified to be combined with WRF-Chem: an observation operator for surface $CO_2$ concentration observations was added to assimilate surface $CO_2$ observations in DART, and some of the code within DART was modified accordingly (Seo and Kim, 2025). The $CO_2$ DA-forecast system used in this study was first developed and comprehensively evaluated in Seo and Kim (2025). Detailed descriptions of the model and data assimilation configurations and extensive analyses of system stability are provided in Seo and Kim (2025).

Final analysis (FNL) (NCEP/NOAA, 2000) was used for the meteorological initial and lateral boundary conditions, and CarbonTracker version 2022 (CT2022) (Jacobson et al., 2023) was used for the chemical lateral boundary conditions. For the chemical initial condition, CT2022 was used only at the first time; thereafter, forecasts were conducted using the analysis produced by assimilating surface $CO_2$ observations as the initial condition. Ensemble forecasts with 20 members were conducted in the $CO_2$ DA cycle, and the multi physical parameterization schemes used in the ensemble forecasts are presented in Table 1.

The experimental period was from June 22 to July 31, 2019, and the spin up period for model stabilization was from June 22 to June 30, 2019. This period was selected to reflect uncertainties from both anthropogenic emissions and biogenic fluxes. A comparison with the 30-year (1991-2020) climatological mean from the ERA5 reanalysis confirmed that the large-scale meteorological conditions in July 2019 were representative of a typical summer in East Asia (not shown). The experimental domain was East Asia with a horizontal resolution of 9 km and 51 vertical layers (Fig. 1). To sufficiently increase the ensemble spread to avoid filter divergence, a 12 h ensemble forecast was conducted only on June 22, 2019, at the beginning of the experiment, and the ensemble DA cycle was conducted every 6 h (00, 06, 12, and 18 UTC) thereafter."

Line 146-162 in the revised manuscript : "The experimental settings for assimilating the surface $CO_2$ concentration using EAKF were as follows: As diurnal cycle of $CO_2$ concentration due to vegetation activities was clear, the assimilation window was ± 1 h and DA cycling was conducted at 6-h intervals. In previous studies that optimized $CO_2$ concentrations and carbon fluxes by assimilating $CO_2$ concentration observations, the cycling interval was 6 h or 24 h (Kang et al., 2011, 2012; Zhang et al., 2021; Liu et al., 2019; Liu et al., 2022). The number of ensembles was set to 20, as preliminary tests showed no significant difference in simulation performance of $CO_2$ concentrations between 20 and 50 members. To prevent spurious correlations, the method of Gaspari and Cohn (1999) was applied for the localization of surface $CO_2$ observations. with a localization radius of 1274.2 km. Initial perturbations were applied to the meteorological and chemical variables of 20 ensemble members to represent initial uncertainties. The initial perturbations of the meteorological variables were obtained from 150 perturbation banks using be.dat.cv3 of the WRF DA (WRFDA). Initial perturbations of the chemical variables (i.e., $CO_2$ concentrations) were produced using the method presented in Yumimoto (2013) and Miao (2014). The lateral boundary conditions for the 20 ensemble members were perturbed using the pert_wrf_bc and update_wrf_bc program in DART. To prevent filter divergence and maintain the ensemble spread during the forecast, both prior and posterior inflation related to model and sampling error, respectively, were applied. Spatially varying state space inflation, based on the Gaussian distribution, was applied as prior inflation (Anderson et al., 2009) and relaxation to prior spread (RTPS) was applied as posterior inflation

(Whitaker and Hamill, 2012). The multiplicative inflation of the RTPS was set to 1.0, which was found to be the most appropriate value in sensitivity tests.”

Line 183 in the revised manuscript : “Self-sensitivity was calculated using analysis at 00, 06, 12, and 18 UTC.”

Line 223-225 in the revised manuscript: “The EFSO was calculated for the 6, 12, 18, and 24 h forecasts at 00 UTC to avoid the influence of the diurnal cycle in $CO_2$ concentrations. To calculate the EFSO, 30 h and 24 h ensemble forecasts were conducted every 18 UTC (corresponding to – 6 h in Fig. 2) and 00 UTC (corresponding to 0 h in Fig. 2), respectively.”

*7. Going into more details in this manuscript raise further questions and concerns, but I limit this review to this list of general issues.*

**Authors' response:** The authors thank the reviewer 2 for a thoughtful review of the manuscript. The reviewer's comments on general issues were very helpful in improving the clarity of the manuscript.

**References**

Casaretto, G., Dillon, M. E., Skabar, Y. G., Ruiz, J. J., and Sacco, M.: Ensemble Forecast Sensitivity to Observations Impact (EFSOI) applied to a regional data assimilation system over south-eastern South America, Atmos. Res., 295, 106996, doi:10.1016/j.atmosres.2023.106996, 2023.

Chang, C. C., Chen, T.-C., Kalnay, E., Da, C., and Mote, S.: Estimating ocean observation impacts on coupled atmosphere-ocean models using Ensemble Forecast Sensitivity to Observation (EFSO), Geophys. Res. Lett., 50(20), e2023GL103154, doi:10.1029/2023GL103154, 2023.

Chen, H. W., Zhang, F., Lauvaux, T., Davis, K. J., Feng, S., Butler, M. P., and Alley, R. B.: Characterization of regional-scale $CO_2$ transport uncertainties in an ensemble with flow-dependent transport errors, Geophys. Res. Lett., 46, 4049-4058, doi:10.1029/2018GL081341, 2019.

Cho, M, and Kim, H. M.: Effect of assimilating $CO_2$ observations in the Korean Peninsula on the inverse modeling to estimate surface $CO_2$ flux over Asia, PLoS One, 17, e0263925, doi:10.1371/journal.pone.0263925, 2022.

Díaz-Isaac, L. I., Lauvaux, T., and Davis, K. J.: Impact of physical parameterizations and initial conditions on simulated atmospheric transport and $CO_2$ mole fractions in the US Midwest, Atmos. Chem. Phys., 18, 14813-14835, doi:10.5194/acp-18-14813-2018, 2018.

Dong, X., Yue, M., Jiang, Y., Hu, X.-M., Ma, Q., Pu, J., and Zhou, G.: Analysis of $CO_2$ spatio-temporal variations in China using a weather-biosphere online coupled model, Atmos. Chem. Phys., 21, 7217-7233, doi:10.5194/acp-21-7217-2021, 2021.

Gasperoni, N. A., Wang, X., Brewster, K. A., and Carr, F. H.: Exploring ensemble forecast sensitivity to observations for a convective-scale data assimilation system over the Dallas-Fort Worth Testbed, Mon. Weather Rev., 152(2), 571-588, doi:10.1175/MWR-D-23-0091.1, 2024.

Gerken, T., Feng, S., Keller, K., Lauvaux, T., DiGangi, J. P., Choi, Y., Baier, B., and Davis, K. J.: Examining $CO_2$ model observation residuals using ACT-America data, J. Geophys. Res.-Atmos., 126, e2020JD034481, doi:10.1029/2020JD034481, 2021.

Hilton, T. W., Davis, K. J., Keller, K., and Urban, N. M.: Improving North American terrestrial $CO_2$ flux diagnosis using spatial structure in land surface model residuals, Biogeosciences, 10(7), 4607-4625, doi:10.5194/bg-10-4607-2013, 2013.

Jung, B.-J., Kim, H. M., Zhang, F., and Wu, C.-C.: Effect of targeted dropsonde observations and best track data on the track forecasts of Typhoon Sinlaku (2008) using an Ensemble Kalman Filter, Tellus Ser. A., 64, 14984, doi:10.3402/tellusa.v64i0.14984, 2012.

Kim, J., Kim, H. M., Cho, C.-H., Boo, K.-O., Jacobson, A. R., Sasakawa, M., Machida, T., Arshinov, M., and Fedoseev, N.: Impact of Siberian observations on the optimization of surface $CO_2$ flux, Atmos. Chem. Phys., 17, 2881-2899, doi:10.5194/acp-17-2881-2017, 2017.

Li, X., Hu, X.-M., Cai, C., Jia, Q., Zhang, Y., Liu, J., Xue, M., Xu, J., Wen, R., and Crowell, S. M. R.: Terrestrial $CO_2$ fluxes, concentrations, sources and budget in Northeast China: Observational and modeling studies, J. Geophys. Res.-Atmos., 125, e2019JD031686, doi:10.1029/2019JD031686, 2020.

Ma, C., Wang, T., Mizzi, A. P., Anderson, J. L., Zhuang, B., Xie, M., and Wu, R.: Multiconstituent data assimilation with WRF-Chem/DART: Potential for adjusting anthropogenic emissions and improving air quality forecasts over eastern China, J. Geophys. Res.-Atmos., 124, 7393-7412, doi:10.1029/2019JD030421, 2019.

Martin, C. R., Zeng, N., Karion, A., Dickerson, R. R., Ren, X., Turpie, B. N., and Weber, K. J.: Evaluation and environmental correction of ambient $CO_2$ measurements from a low-cost NDIR sensor, Atmos. Meas. Tech., 10, 2383-2395, doi:10.5194/amt-10-2383-2017, 2017.

Peng, Z., Kou, X., Zhang, M., Lei, L., Miao, S., Wang, H., Jiang, F., Han, X., and Fang, S.: $CO_2$ flux inversion with a regional joint data assimilation system based on CMAQ, EnKS, and surface observations, J. Geophys. Res.-Atmos., 128, e2022JD037154, doi:10.1029/2022JD037154, 2023.

Raeder, K., Anderson, J. L., Collins, N., Hoar, T. J., Kay, J. E., Lauritzen, P. H., and Pincus, R.: DART/CAM: An ensemble data assimilation system for CESM atmospheric models, J. Clim., 25(18), 6304-6317, doi:10.1175/JCLI-D-11-00395.1, 2012.

Seo, M.-G., and Kim, H. M.: Effect of meteorological data assimilation using 3DVAR on high-resolution simulations of atmospheric $CO_2$ concentrations in East Asia, Atmos. Pollut. Res., 14(6), 101759, doi:10.1016/j.apr.2023.101759, 2023.

Seo, M.-G., Kim, H. M., and Kim, D.-H.: Effect of atmospheric conditions and VPRM parameters on high-resolution regional $CO_2$ simulations over East Asia, Theor. Appl. Climatol., 155, 859-877, doi:10.1007/s00704-023-04663-2, 2024a.

Seo, M.-G, Kim, H. M., and Kim, D.-H.: High-resolution atmospheric $CO_2$ concentration data simulated in WRF-Chem over East Asia for 10 years, Geosci. Data J., 11, 1024-1043, doi:10.1002/gdj3.273, 2024b.

Seo, M.-G., and Kim, H. M.: Evaluation of high-resolution regional $CO_2$ data assimilation-forecast system in East Asia using observing system simulation experiment and effect of observation network on simulated $CO_2$ concentrations, Q. J. R. Meteorol. Soc., e4987, doi:10.1002/qj.4987, 2025.

Shusterman, A. A., Kim, J., Lieschke, K. J., Newman, C., Wooldridge, P. J., and Cohen, R. C.: Observing local $CO_2$ sources using low-cost, near-surface urban monitors, Atmos. Chem. Phys., 18, 13773-13785, doi:10.5194/acp-18-13773-2018, 2018.

Stull, R. B.: An introduction to boundary layer meteorology, Vol. 13: Springer Science & Business Media, 1988.

Yang, E.-G., Kim, H. M., Kim, J., and Kay, J. K.: Effect of observation network design on meteorological forecasts of Asian dust events, Mon. Weather Rev., 142, 4679-4695,

doi:10.1175/MWR-D-14-00080.1, 2014.

Zheng, T., French, N. H. F., and Baxter, M.: Development of the WRF-CO$_2$ 4D-Var assimilation system v1.0, Geosci. Model Dev., 11, 1725-1752, doi:10.5194/gmd-11-1725-2018, 2018.